# Heavy metals in water and sediment of Cikijing River, Rancaekek District, West Java: Contamination distribution and ecological risk assessment

**Mariana Marselina**[ID]*, **M. Wijaya**\*

Environmental Engineering Study Program, Faculty of Civil and Environmental Engineering, Bandung Institute of Technology Jl. West Java, Indonesia

\* mariana.marselina@yahoo.com (MM); m.wijaya@students.itb.ac.id (MW)

**Data Availability Statement:** All relevant data are within the paper and its Supporting Information files.

**Funding:** All authors would like to thank Bandung Institute of Technology for funding this research

## Abstract

The Cikijing River is one of the rivers of the Citarik River Basin, which empties into the Citarum River and crosses Bandung Regency and Sumedang Regency, Indonesia. One of the uses of the Cikijing River is as a source of irrigation for rice fields in the Rancaekek area, but the current condition of the water quality of the Cikijing river has decreased, mainly due to the disposal of wastewater from the Rancaekek industrial area which is dominated by industry in the textile and textile products sector. This study aims to determine the potential ecological risks and water quality of the Cikijing River based on the content of heavy metals (Cr, Cu, Pb, and Zn). Sampling was carried out twice, during the dry and rainy seasons at ten different locations. The selection of locations took into account the ease of sampling and distribution of land use. Based on the results of this study, it was found that the water quality of the Cikijing River was classified as good based on the content of heavy metals (Cr, Cu, Pb, and Zn) with a Pollution Index 0.272 (rainy season) and 0.196 (dry season), while for the sediment compartment of the Cikijing River, according to the geoaccumulation index ($I_{geo}$) were categorized as unpolluted for heavy metals in rainy and dry seasons Cr (-3.16 and -6.97) < Cu (-0.59 and -1.05), and Pb (-1.68 and -1.91), heavily to very heavily polluted for heavy metals Zn (4.7 and 4.1) . The pollution load index (PLI) shows that the Cikijing River is classified as polluted by several heavy metals with the largest pollution being Zn> Cu > Pb > Cr. Furthermore, the results of the analysis using the Potential Ecological Risk Index (PERI) concluded that the Cikijing River has a mild ecological risk potential in rainy season (93.94) and dry season (96.49). The correlation test results concluded that there was a strong and significant relationship between the concentrations of heavy metals Pb and Zn and total dissolved solids, salinity, and electrical conductivity in the water compartment.

especially Research, Community Service, and Innovation Program of 2023 (P2MI 2023)

**Competing interests:** The authors have declared that no competing interests exist.

## 1 Introduction

Heavy metal pollution into rivers through various sources, both natural and anthropogenic, is of great concern because it is a serious environmental threat to living organisms and aquatic ecosystems as well as its non-biodegradability, bioaccumulation, environmental stability, persistence, and biotoxicity [1]. Heavy metals in rivers over time will precipitate on the riverbed so that the concentration of heavy metals in river sediments will be higher than in water; this is because heavy metals tend to bind to hydroxide and organic matter in sediments [2].

Heavy metals are toxic and will accumulate in the environment thus impacting humans. In waters, heavy metals can accumulate in sediments and aquatic biota through the processes of bioconcentration, bioaccumulation, and biomagnification. Some heavy metals such as Arsenic (As), Lead (Pb), Mercury (Hg), and Cadmium (Cd) have toxic properties to humans. The toxic effects that appear on organs and tissues of the human body are the result of the interaction of heavy metals with important cell molecules, thereby damaging the structure and function of target organ cells [3–5].

Ustaoglu et. al (2019) [6] have done an important study regarding spatial distribution and variation of heavy metals in Pazarsuyu Stream, Turkey to identified drinking water safety in Bulancak city. In that study, the heavy metal contamination in the sediments was evaluated by applying the enrichment factor (EF), contamination factor (CF), geoaccumulation index (Igeo) and potential ecological risk index (PERI). The results show that Pb have a significant enrichment of contamination according to calculated EF values. Therefore, the stream water can be used for irrigation but extensive treatment required before using for domestic purposes to prevent public health effects. Similar study also have done by Ustaogle et.al (2020) [7] to asses sediment pollution by heavy metals fro, Akcaova Stream and Calislar Stream and the result suggested that metal contamination sourced most likely from the naturally present minerals in the area.

In the other hand Tepe et.al (2022) [8] have done the assesment of potential contamination levels and human health risk of heavy metals in sediment of the Turnasuyu Stream in Ordu, Turkey. The results show that all heavymetal levels, except Cd and Pb, were in the minimum enrichment range as assessed by the sediment enrichment factor (EF). The low risk of the study area has also been confirmed by the ecological risk index (Eri) values and the probable human health risk assessment has been performed by lifetime cancer risk (LCR) values. The result also show that the hazard index (HI) and total hazard index (THI) results were both higher in children than in adults. Another study by Khalijian A. et al., (2022) [9] also have done ecotoxicological Assessment of Potentially Toxic Elements (As, Cd, Ni and V) in the Sediments of Southern Part of Caspian Sea, Iran Province by geoaccumulation index (Igeo) values. It is found that that the levels of contamination were higher in the areas where industrial, domestic and agricultural wastewater was discharged.

Heavy metals in surface water adsorb and settle into sediments through a series of physical and chemical reactions and then return to the water column in the form of a dissolved state through complex hydrodynamic action. This effect will also constantly be retained in environmental media and accumulate in aquatic organisms such as into macrophytes and the other water plantation, then consequently, the heavy metals will be transmitted to humans through the food chain [10]. Those process of element transportation can be shown in several related studies. Rakib et.al have done the assessment of trace element toxicity ih Halda River, Bangladesh. The result show that the levels of trace elements in water samples were much higher than the guideline values for safe limits of drinking water and the result also indicated that water from this river is not safe for drinking purposes [11]. Contamination in sediments have shown by Ali et al.(2022) [12] in Bhairab River, Bangladesh. The result show that the contamination

level of toxic metals in sediment show that the condition is frightening and probably severely affecting the aquatic ecology of this riverine ecosystem. Accumulation in water plantation have done by Hosain et.al to assess eight metals, Fe, Zn, Mn, Cu, Co, Rb, Sr, and Pb in Porteresia sp. from the six salt marsh sites of Bangladesh. The results show that the roots of the Porteresia sp. showed high accumulation of the metals when compared to shoots and leaves suggesting relevant availability in the sediment. Pb was the only metal with concentrations significantly higher in the leaves and shoots than in the root [13].

Heavy metals have harmful effects on human health, and exposure to these elements has been increased by industrial or modern industrialization. Contamination of water and air by toxic metals is an environmental concern and hundreds of millions of people are being affected in all over the world. Nowadays, food contamination with heavy metals is also another concern for animal health. Sobhanardakani (2022) studied about the health risk assessment of the heavy metals in caviar of wild Persian sturgeon. It is found that the contents of Ba, Cr, Fe, Hg, Mn and Zn (mg kg− 1) in caviar samples were 0.95, 0.27, 71.3, 1.44, 0.01 and 17.0, respectively. Based on this results, monitoring of chemicals accumulation in the foodstuff is recommended since the concentration of Fe and Hg were higher than Maximum Permissible Limits in the caviar samples [14].

Effects of heavy metal not only harms animal health but also threaten plant species. Hosseini et. al (2020) assessed the content of heavy metal in such plant species along some highways in Hamedan, West of Iran. The results found that the mean contents of elements (mg kg −1) in aerial parts of Achillea wilhelmsii were 0.16 for Cd, 4.52 for Cu, 1.91 for Pb, 1.70 for Ni, and 44.80 for Zn, while in the aerial part samples of Cardaria draba, the concentrations (mg kg −1) and the mean contents were 0.16, 2.29, 2.58, 1.60, and 31.29, respectively. This meant that the traffic volume affected the contents of the metals in the soil and the herbaceous species. The metal content in herbaceous tissues varied significantly between plant species [15].

There are several studies that have been done in some area in Indonesia regarding heavy metal contamination. For example, in Bandung Regency area, the rice fields have been contaminated by heavy metals of Pb, Cd, Cr, and Ni [16]. Another study on pollution in agricultural land was conducted in Nganjuk Regency. Based on the results of this study, it was found that the value of Cd is higher than other concentrations, and this has a significant impact on population health [17].

The Cikijing River is one of the rivers used as a source of water for irrigating rice fields in the Rancaekek area. However, the current condition of the Cikijing river water quality has decreased, mainly due to the disposal of liquid waste from the textile and textile product industries. Previous research found that Cikijing river water was exposed to relatively high concentrations of heavy metals, mainly from dyes in textile industry wastewater. The heavy metal detected with the highest concentration was chromium (Cr) at 0.3026 mg/L, and the concentration of zinc (Zn) reached 2.922 mg/L [18]. Other research found that lead (Pb) was detected in the Cikijing River water body with a concentration of 1.38 mg/L and Cu was detected with a concentration of 0.57 mg/L [19]. The existence of heavy metal pollution in the Cikijing river will not only pollute the river and the ecosystem in it but also pollute the surrounding agricultural land, considering that the river water is also used as a source of irrigation. The presence of heavy metal pollution on agricultural land results in the inhibition of the rate of photosynthesis, changes in cell shape, and inhibits the rate of ecosystem growth [20].

Based on research, the Soil in the Cikijing River Watering Area, Rancaekek District was identified that the heavy metal Cu content was above the soil quality standard. The highest total concentration of Cu and Zn metals in the soil was found in Linggar village, which was 91.90 mg/Kg and 483.43 mg/Kg [21]. The highest concentration of available Cr metal in the soil and above the critical limit is found in paddy fields in Linggar Village, which is 2.96 mg/

Kg because Linggar Village is the location of a textile factory. The highest concentration of available Pb in the soil was found in paddy fields in Bojongloa village, namely 0.70 mg/Kg. The high concentration of available Pb in the soil in Bojongloa Village is due to the proximity of the sampling location to housing, this can make the sample location contaminated with household waste so that the heavy metal available Pb content at this location is high [22].

In previous studies, there was no analysis and discussion regarding the potential dangers of ecological risks that could arise. Therefore, research studies on heavy metals are important considering that the Cikijing river contributes to peoples' daily lives. To assess how much heavy metal pollution is in water and to provide input to improve quality if there is a decrease in quality due to the presence of pollutant compounds, the Pollution Index (PI) method is used. Meanwhile, three assessment methods were used for sediments, namely the Potential Ecological Risk Index (PERI), to assess the extent of the ecological hazard caused by heavy metal contamination. In addition, for the assessment of heavy metals in aquatic sediments, the geoaccumulation index ($I_{geo}$) model is also used to determine the level of heavy metal pollution in sediments by considering the effects of variations in the layers of the earth and the Pollution Load Index (PLI) which is useful for determining the status of heavy metal pollution in sediments in the water.

Based on the background above, this study aims to assess the water quality and measure the potential ecological risks that can be caused by heavy metal pollution (Cr, Cu, Pb, and Zn) in the Cikijing River. In addition, this study also aims to analyze the relationship between the physical-chemical parameters of the Cikijing River water (pH, temperature, dissolved oxygen, total dissolved solids (TDS), total suspended solids (TSS), salinity, and electrical conductivity) and the concentration of heavy metals (Cr, Cu, Pb, and Zn) in the water compartment, and analyze the relationship between organic matter content and sediment texture (sand and silt) and heavy metal concentrations (Cr, Cu, Pb, and Zn) in the sediment compartment.

## 2 Materials and methods

### 2.1 Location and time of sampling

The research in the form of field data collection, water sampling, and sediment sampling was carried out once during the rainy season, namely on 23–24 March 2022, and once during the dry season, namely on 29 June 2022. This research was conducted on the body of the Cikijing River that crosses two administrative areas, namely the Cimanggung District, Sumedang Regency and the Rancaekek District, Bandung Regency, West Java Province. In this study, there was no need for a permit from the relevant authority because the Cikijing River, both upstream and downstream, is a public facility and is not a vital asset belonging to a private company. Determination of the sampling point uses the purposive sampling method, namely the sampling point will be determined deliberately with reference to the Indonesian National Standard (SNI) 6989.57: 2008 concerning Water and Wastewater Section 57: Surface Water Sampling Methods.

Sampling of water and sediment of the Cikijing River was carried out at ten stations by utilizing crossing bridges made of either bamboo or concrete. The determination of these ten stations is based on land use in the vicinity, namely stations 1 and 2 are upstream areas and community settlements. Station 3 is the location where the discharge flow from the industrial wastewater treatment plant has been identified. Stations 4, 5, and 6 are agricultural land, while stations 7–10 are community settlements. In addition, determining the location of river water sampling is based on the following points.

a. Upstream areas or natural source water areas, namely locations where pollution has not occurred.

b. River water utilization or allotment area, namely a location where river water will be used as raw material for drinking water, water for recreation, industry, fisheries, agriculture and others.

c. Areas that are potential recipients of contaminants, namely locations that experience changes in water quality due to industrial, agricultural, domestic activities and so on.

d. The area where two rivers meet or where a tributary enters.

Geographically, the research location is in Sawahdadap Village, Cimanggung District for stations 1 and 2. While stations 3–6 are in Linggar Village and stations 7–10 are in the administrative area of the Jelegong Village, both of which are included in the Rancaekek District. The map of the research location can be seen in Fig 1.

## 2.2 Research materials and equipment

The materials used in the field in this study were water samples, sediment samples, $HNO_3$ solution which was used as a preservative in water samples for heavy metal testing in the laboratory, aquadest as a material for calibrating research equipment, and blue ice which was used

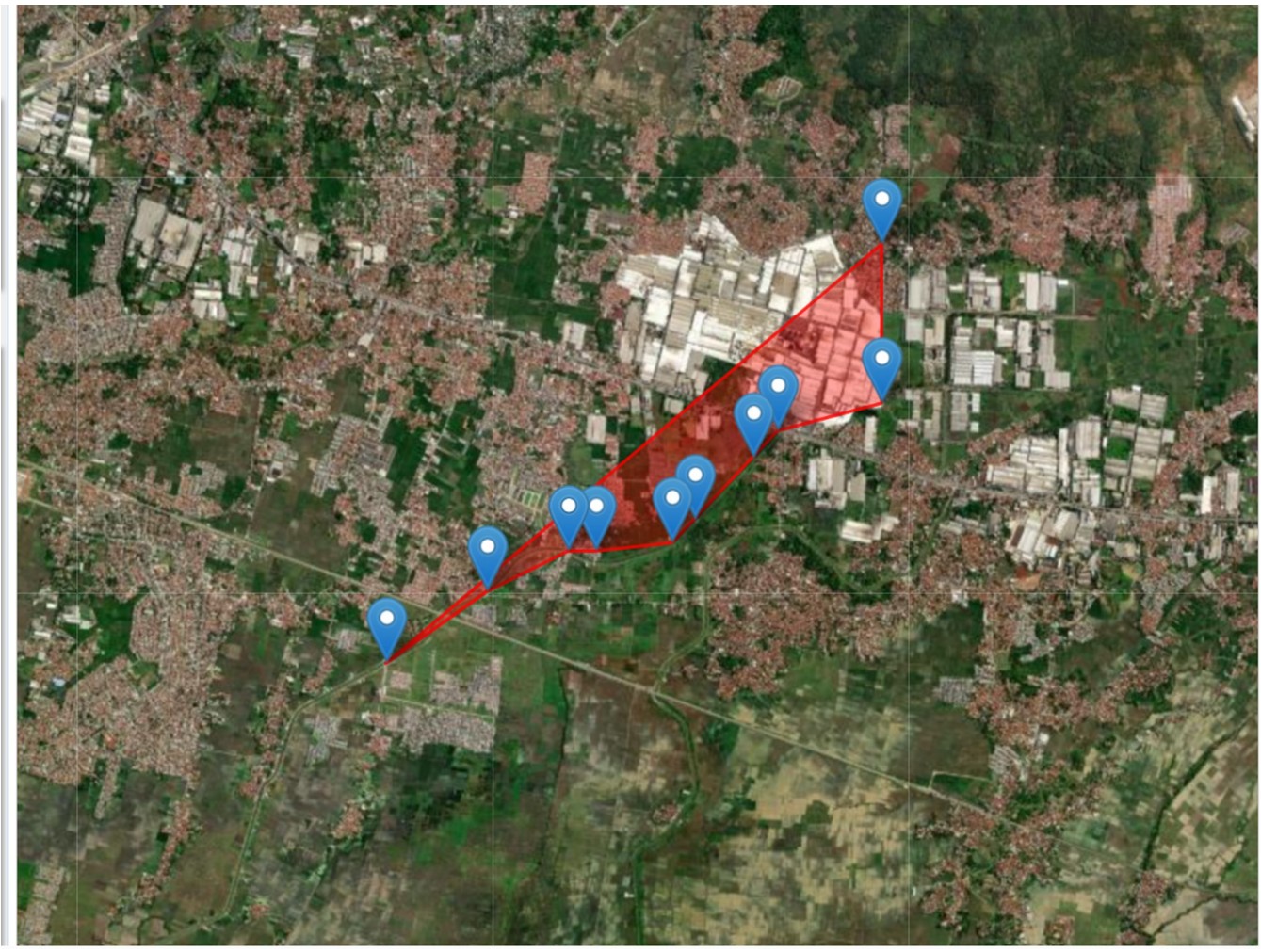

**Fig 1. Map of sampling stations the Cikijing River.** Spatial information were obtained from http://eros.usgs.gov/#.

to preserve water samples and sediment during the journey from the sampling location to the laboratory. In this study there were also measurements of the physical-chemical parameters in the Cikijing River. The equipment used in field research included high-density Polyethylene (HDPE) plastic bottles, cool boxes, markers, water samplers, pH meters, dissolved oxygen (DO) meters, dippers, Ekman grab samplers, buckets, and small shovels.

## 2.3 Data collection

Two forms of data collection were employed in this study: direct data testing in the field (in situ) and data testing conducted in the laboratory (ex situ). The parameters to be measured directly in the field were the physical-chemical parameters of river water, in the form of pH, temperature, and dissolved oxygen (DO) which aims to determine the condition of the Cikijing river water when sampling. Data testing in the laboratory was carried out for water and sediment samples from the Cikijing River. Water and sediment samples were tested in the laboratory to measure heavy metal concentrations (Cr, Cu, Pb, and Zn). In addition, physical-chemical parameters were measured in the laboratory in the form of total dissolved solids, total suspended solids, salinity, and electrical conductivity.

Based on data, the water discharge of the downstream Cikijing River is 0.399–1.29 m3/second.. Therefore, water discharge below 5 $m^3$/second will be sampled at a point in the middle of the river at a depth of 0.5 times the depth of the surface. Sampling was carried out twice where in one sampling time, a total of 10 water samples and 10 sediment samples. Therefore, the total samples in this study were 20 water samples and 20 sediment samples.

The process of testing for heavy metal content in water and sediment samples was carried out based on the Indonesian National Standard (SNI) for each type of heavy metal tested. The reference standards applied for testing heavy metals in Cikijing River water samples refer to SNI No. 6989.6–2009 for Cu metal, SNI No. 6989.7–2009 for Zn metal, SNI No. 6989.8–2009 for Pb metal, and SNI No. 6989.8–2009 for Cr metal. Meanwhile, heavy metal testing in the Cikijing River sediment samples, it refers to SNI No. 8910:2021. Measuring the content of heavy metals in the sample using the atomic absorption Spectrophotometry (AAS) tool, the sediment sample destruction process was carried out first. The function of destruction is to break the bonds between organic compounds and the metal to be analyzed. The destruction process was carried out to decompose the metal compounds into inorganic metal forms or to break down compounds into their elements so that they can be analyzed [18].

Heavy Metal Measurement, first weighing of the sediment sample that has been homogenized, Add 10 mL $HNO_3$ which serves as digestion, Heat the test sample solution at 95˚C ± 5˚C for 10 minutes– 15 minutes without boiling, Add 5 mL concentrated $HNO_3$, cover again with a watch glass and reheat the test sample at 95˚C ± 5˚C for 30 minutes. If the smoke is brown and the solution is still cloudy, add 5 mL of concentrated $HNO_3$ again and repeat heating until the solution is clear and/or the brown smoke disappears, then add 2 mL of mineral-free water and 3 mL of 30% $H_2O_2$ to create a peroxide reaction, then add continuously. gradually 1 mL of 30% $H_2O_2$ until the foam is reduced or the test sample does not change, Add 10 mL of concentrated HCl to the test sample according to the steps and continue heating until the volume of the test sample solution reaches 5 mL or heat the test sample at 95˚C ± 5˚C without boiling for 15 minutes, then cool, Filter the sample solution and collect the filtrate in a 100 mL volumetric flask, Samples are ready to be measured for absorption using atomic absorption spectroscopy.

The sediment samples obtained were then tested for sediment texture using a sieving method (sieve prior) based on Buchanan (1971) [22] and separating the texture into two fractions, namely sand and silt. Testing for organic matter in sediments used the gravimetric

method based on Sudjadi (1971) [23]. The gravimetric method used a furnace where the temperature reached 550°C ± 5°C in oven for four hours.

## 2.4 Data analysis method

**2.4.1 Water quality analysis method.** The results of measurements of heavy metal concentrations (Cr, Cu, Pb, and Zn) in Cikijing River water were analyzed using the pollution index (PI), which is a method for determining water quality status as stipulated in the Decree of the Minister of Environment Number 115 of 2003. The water pollution index is calculated using Eq (1) as follows.

$$PI_j = \sqrt{\frac{\left(\frac{c_i}{L_{ij}}\right)^2_M + \left(\frac{c_i}{L_{ij}}\right)^2_R}{2}} \tag{1}$$

where *PIj* is the pollution index for the allotment (j), which is a function of *Ci/Lij*, *Ci* denotes the concentration of water quality parameter i, and *Lij* denotes the concentration of water quality parameter *i* included in the quality standard for water allotment *j*, while *M* represents the maximum, and *R* represents the average. In this case, the PI index class consists of 4 (four) with the criteria shown in Table 1.

**2.4.2 Methods of analysis of heavy metal distribution in water and sediments.** The distribution or mobility of heavy metals in the Cikijing River body was determined by calculating the organic-carbon ($K_{oc}$) and Octanol-water ($K_{ow}$) coefficients. The organic-carbon coefficient ($K_{oc}$) is calculated using Eq (2) as follows.

$$K_{oc} = \frac{K_D}{f_{oc}} \tag{2}$$

The $K_D$ value is the water-sediment partition coefficient ($K_D = [C]_{sed} / [C]_{water}$) while $f_{oc}$ is the fraction of organic carbon in the sediment (%). However, this study only measured the percentage of total organic matter in the sediment, so the value of the organic carbon fraction used a value of 58% of the total organic matter [24]. The level of heavy metal mobility based on the organic carbon fraction value is shown in Table 2.

The octanol-water coefficient ($K_{ow}$) calculation describes the hydrophobic or hydrophilic properties of a compound. Types of Heavy Metal Compounds based on Log $K_{ow}$ Values can be seen in Table 3. The equation of the octanol-water ($K_{ow}$) coefficient is as follows [26].

$$Log\ K_{oc} = 0.903 \log K_{ow} + 0.094 \tag{3}$$

**2.4.3 Ecological risk potential analysis method.** The results of measuring the concentration of heavy metals (Cr, Cu, Pb, and Zn) in the Cikijing River sediments were analyzed using three assessment methods, as follows.

a. Geoaccumulation Index ($I_{geo}$)
   The $I_{geo}$ value is obtained using Eq (4) listed below.

$$I_{geo} = log_2\left(\frac{M_c}{1,5 \times B_c}\right) \tag{4}$$

**Table 1. Water quality category based on pollution index value.**

| PI value | 0 < PI < 1.0 | 1.0 < PI < 5.0 | 5.0 < PI < 10.0 | PI > 10.0 |
|---|---|---|---|---|
| Category | meets quality standards | lightly polluted | moderately polluted | heavily polluted |

**Table 2. Heavy metal mobility level based on Log Koc value [25].**

| Log Value $K_{oc}$ | Category |
|---|---|
| <1.176 | Very Easy to Move |
| 1.176–1.875 | Easy to Move |
| 1.875–2.698 | Easy enough to Move |
| 2.698–3.6 | Move a little |
| >3.6 | Not Moving |

The $M_c$ value is the concentration of a heavy metal measured in the sediment sample and the $B_c$ value is the natural concentration of the heavy metal. The factor 1.5 in Eq (4) is a correction factor for natural fluctuations related to lithospheric effects (considering the effects of variations in the earth's layers) [28].

b. Pollution Load Index (PLI)

Before getting the PLI value, the contamination factor (CF) value of each heavy metal being analyzed must first be calculated. This contamination factor (*CF*) is the ratio between the concentration of a heavy metal in a sediment sample and its natural concentration (background concentration) or permissible heavy metal quality standards. The CF value is used to determine the level of water pollution for a type of metal in the sediment is low, moderate, high, or very high [29]. The CF and PLI values are calculated using Eqs (5) and (6) as follows.

$$CF = \frac{M_c}{B_c} \tag{5}$$

$$PLI = \left(CF_1 \times CF_2 \times CF_3 \times \ldots \times CF_n\right)^{\frac{1}{n}} \tag{6}$$

The $M_c$ value is the concentration of a heavy metal measured in the sediment sample and the $B_c$ value is the natural concentration of the heavy metal. The value of $n$ is the number of types of heavy metals analyzed at measurement point.

c. *Potential Ecological Risk Index*(PERI)

The PERI value calculation is obtained by Eqs (7) and (8) as follows.

$$C_f^i = \frac{C_{0-1}^i}{C_n^i} \tag{7}$$

$C_f^i$ is the measured concentration level of heavy metals in sediment samples, which is a

**Table 3. Types of heavy metal compounds based on Log $K_{ow}$ values [27].**

| $K_{ow}$ Log Value | Category |
|---|---|
| >**0** | hydrophobic compounds (low solubility in water) |
| <**0** | hydrophilic compound (high solubility in water) |

reference or quality standard for ecological risk of heavy metals in sediments $C_{0-1}^i C_n^i$.

$$PERI = \sum_{i=1}^{n} E_r^i = \sum_{i=1}^{n} T_r^i \cdot C_r^i \tag{8}$$

$E_r^i$ is the index of ecological potential of a certain heavy metal $i$ which is the toxicity response factor for heavy metal $i$, and PERI is the index of potential ecological risk which is the sum of $T_r^i E_r^i$. Categories of classification of pollution due to heavy metals in sediments in a waters based on the Geoaccumulation Index, PLI, and PERI can be seen in Tables 4 and 5.

**2.4.4 Statistical analysis methods.** Statistical analysis carried out in this study was a correlation analysis between data on heavy metal concentrations (Cr, Cu, Pb, and Zn) measured in water and sediment and the physical-chemical parameters of Cikijing River water, organic matter, and sediment texture (sand and silt) using the *IBM SPSS Statistics 25 software*. If (sig. $< 0.05$) then a non-parametric analysis is used because the pollutant concentration data is not normally distributed, whereas if the data is normally distributed (sig. $> 0.05$), then a parametric analysis is used.

# 3 Results and discussion

## 3.1 Water quality based on heavy metals in the Cikijing River

The concentration of heavy metals for Lead (Pb), Chromium (Cr), Zinc (Zn), and Copper (Cu) types contained in the Cikijing river water is used in assessing the quality of the river. The Cikijing River based on the Decree of the Governor of West Java No. 39 of 2000 designated as a source of agricultural or irrigation water. Therefore, the assessment of the water quality of the Cikijing River refers to Government Regulation of The Republic of Indonesia No. 22 of 2021 concerning the Implementation of Environmental Protection and Management.

As seen in Table 6 the measurements of the majority of heavy metal concentrations (Cr and Cu) at ten measurement points both in the rainy and dry seasons were below the detection limit of the atomic absorption spectrophotometry (AAS) measuring instrument. Furthermore, the data from the concentration measurements were analyzed by comparison between the average concentration of heavy metals analyzed and the standards for water quality; it was found that the concentrations of heavy metals Pb, Cr, Zn, and Cu did not exceed the quality standards according to the designation of the Cikijing River as a source of agricultural irrigation. Reviewing the Decree of the Minister of Environment No. 115 of 2003 in article 2 paragraph 1, determining the status of water quality can be analyzed using the Pollution Index (PI) method. By adhering to this standard, it is hoped that water quality management on the basis

**Table 4. Categories of heavy metal pollution in waters based on $I_{geo}$ [30] and PLI values [29].**

| $I_{geo}$ Value | Pollution level | CF and PLI values | Pollution Category |
|---|---|---|---|
| $I_{geo} \leq 0$ | Unpolluted | CF<1 | low pollution by certain metals |
| $0 < I_{geo} \leq 1$ | unpolluted to moderately polluted | 1<CF<3 | moderate pollution by certain metals |
| $1 < I_{geo} \leq 2$ | moderately polluted | 3<CF<6 | high pollution by certain metals |
| $2 < I_{geo} \leq 3$ | moderately to heavily polluted | CF>6 | heavy pollution polluted by certain metals |
| $3 < I_{geo} \leq 4$ | heavily polluted | | |
| $4 < I_{geo} \leq 5$ | heavily polluted to very heavily polluted | PLI<1 | not polluted by a combination of several metals |
| $I_{geo} > 5$ | very heavily polluted | PLI > 1 | polluted by a combination of several metals |

**Table 5. Factors of heavy metal toxic response and ecological risk categories of heavy metals in waters based on PERI values [31].**

| Heavy metal type | Heavy metal toxic response factor (T) | PERI value | Level of potential ecological risk |
|---|---|---|---|
| Cr | 2 | PERI<150 | mild ecological risk |
| Cu | 5 | 150≤PERI<300 | moderate ecological risk |
| Pb | 5 | 300≤PERI<600 | bad ecological risk |
| Zn | 1 | PERI≥600 | very bad ecological risk |

of the Pollution Index can provide input to decision makers in order to be able to assess the quality of water bodies for a designation and to take action to improve quality if there is a decrease in water quality due to the presence of pollutant compounds. One other alternative is to use the water quality index (WQI), but the use of the water quality index with water quality index (WQI) is not effective because many important parameters are not present in the formula, such as nutrients, heavy metals and coliform.

Table 7 shows that the metals Cr, Cu, Pb, and Zn have met the quality standards with the acquisition of PI values in the range of 0–1 . Based on the four types of heavy metals analyzed in water bodies, the average PI value for the Cikijing River is classified as unpolluted.

## 3.2 Analysis of heavy metal distribution in water and sediment compartments

The concentration of heavy metals Cr, Cu, Pb, and Zn in sediments has a much greater value than in the water compartment. In both sediment and water, the heavy metal Zn has the highest concentration when compared to other types of heavy metals. However, but the highest level of contamination only occurs in sediments because Zn metal in water is still classified as a quality standard. These results shown in Figs 2–4.

**Table 6. Results of measurement of heavy metal concentrations (Cr, Cu, Pb, and Zn) in water compartment.**

| Sampling Point | Rainy Season (23–24 March 2022) | | | | Dry Season (29 June 2022) | | | |
|---|---|---|---|---|---|---|---|---|
| | Cr (mg/L) | Cu (mg/L) | Pb (mg/L) | Zn (mg/L | Cr (mg/L) | Cu (mg/L) | Pb (mg/L) | Zn (mg/L) |
| 1 | 0.007 | <0.024 | <0.002 | 0.002 | <0.210 | <0.060 | <0.001 | 0.002 |
| 2 | 0.011 | <0.031 | <0.001 | 0 | <0.222 | <0.062 | <0.008 | <0.002 |
| 3 | <0.005 | <0.028 | 0.002 | 0.015 | <0.173 | <0.055 | 0.015 | 0.039 |
| 4 | <0.005 | <0.025 | 0.002 | 0.769 | <0.177 | <0.060 | 0.055 | 0.038 |
| 5 | <0.005 | <0.026 | 0.002 | 0.037 | <0.195 | <0.054 | 0.018 | 0.041 |
| 6 | <0.005 | <0.031 | 0.003 | 0.048 | <0.289 | <0.054 | 0.023 | 0.040 |
| 7 | <0.004 | <0.010 | 0.003 | 0.002 | <0.272 | <0.063 | 0.021 | 0.027 |
| 8 | <0.005 | <0.015 | 0.003 | 0.033 | <0.212 | <0.069 | 0.036 | 0.028 |
| 9 | <0.004 | <0.028 | 0.003 | 0.004 | <0.223 | <0.069 | 0.021 | 0.026 |
| 10 | <0.005 | <0.020 | 0.003 | 0.058 | <0.248 | <0.020 | 0.022 | 0.031 |
| **Maximum** | 0.011 | 0.000 | 0.003 | 0.769 | 0.000 | 0.000 | 0.055 | 0.041 |
| **Minimum** | 0.007 | 0.000 | 0.002 | 0.000 | 0.000 | 0.000 | 0.015 | 0.002 |
| **Average** | 0.006 | 0.024 | 0.002 | 0.097 | 0.222 | 0.057 | 0.022 | 0.027 |
| **SD** | 0.002 | 0.007 | 0.001 | 0.237 | 0.038 | 0.014 | 0.015 | 0.015 |
| **CV(%)** | 36.9 | 29.0 | 29.1 | 245 | 17.2 | 24.7 | 67.6 | 53.0 |
| **Quality Standard (IV)** | 1.0 | 0.5 | 0.2 | 2.0 | 1.0 | 0.5 | 0.2 | 2.0 |
| **Category** | Under Quality standards | Under Quality Standard | Under Quality Standard | Under Quality Standard | Under Quality Standard | Under Quality Standard | Under Quality Standard | Under Quality Standard |

**Table 7. Pollution index results based on heavy metals Cr, Cu, Pb, and Zn in the Cikijing River in the measurement of the rainy and dry seasons.**

| | POLLUTION INDEX | | | | | | | |
|---|---|---|---|---|---|---|---|---|
| | Rainy Season (*Ci/Lij*) | | | | Dry Season (*Ci/Lij*) | | | |
| Parameter | cr | Cu | Pb | Zn | cr | Cu | Pb | Zn |
| Maximum | 0.011 | 0 | 0.017 | 0.384 | 0 | 0 | 0.275 | 0.020 |
| Average | 0.002 | 0 | 0.011 | 0.048 | 0 | 0 | 0.106 | 0.014 |
| **PI** | 0.272 | | | | 0.196 | | | |
| **Category** | Meets the Quality Standards (Good Water Quality) | | | | Meets the Quality Standards (Good Water Quality) | | | |

The high contamination in sediments from heavy metals Cr, Cu, Pb, and Zn is influenced by the density of these heavy metals based on Stokes' law (vertical deposition) [32]. The density values of Pb, Cu, Cr, and Zn are 11.34 g/cm$^3$; 8.96 g/cm$^3$; 7.19 g/cm$^3$; and 7.14 g/cm$^3$, respectively. In addition, another influence on the deposition process is the water current velocity which results in a particle experiencing a deposition process that is not in accordance with the initial point of entry of these heavy metals into the river body. Both of these forces have a term in the form of drag force, which is the resultant of the vertical force and the horizontal force due to the influence of the gravitational force and current speed.

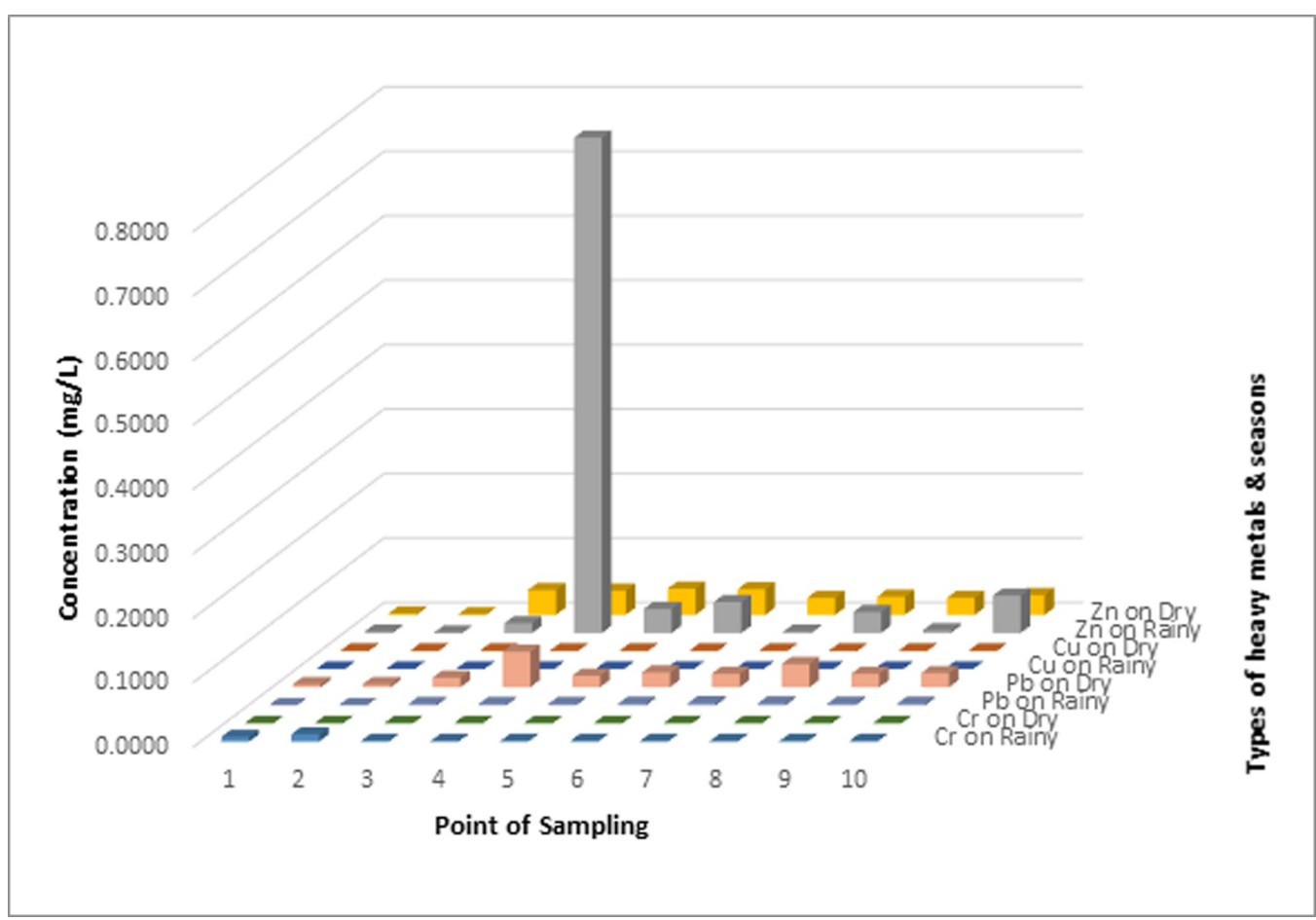

**Fig 2. Comparison of heavy metal concentrations in water compartments in the rainy and dry seasons.**

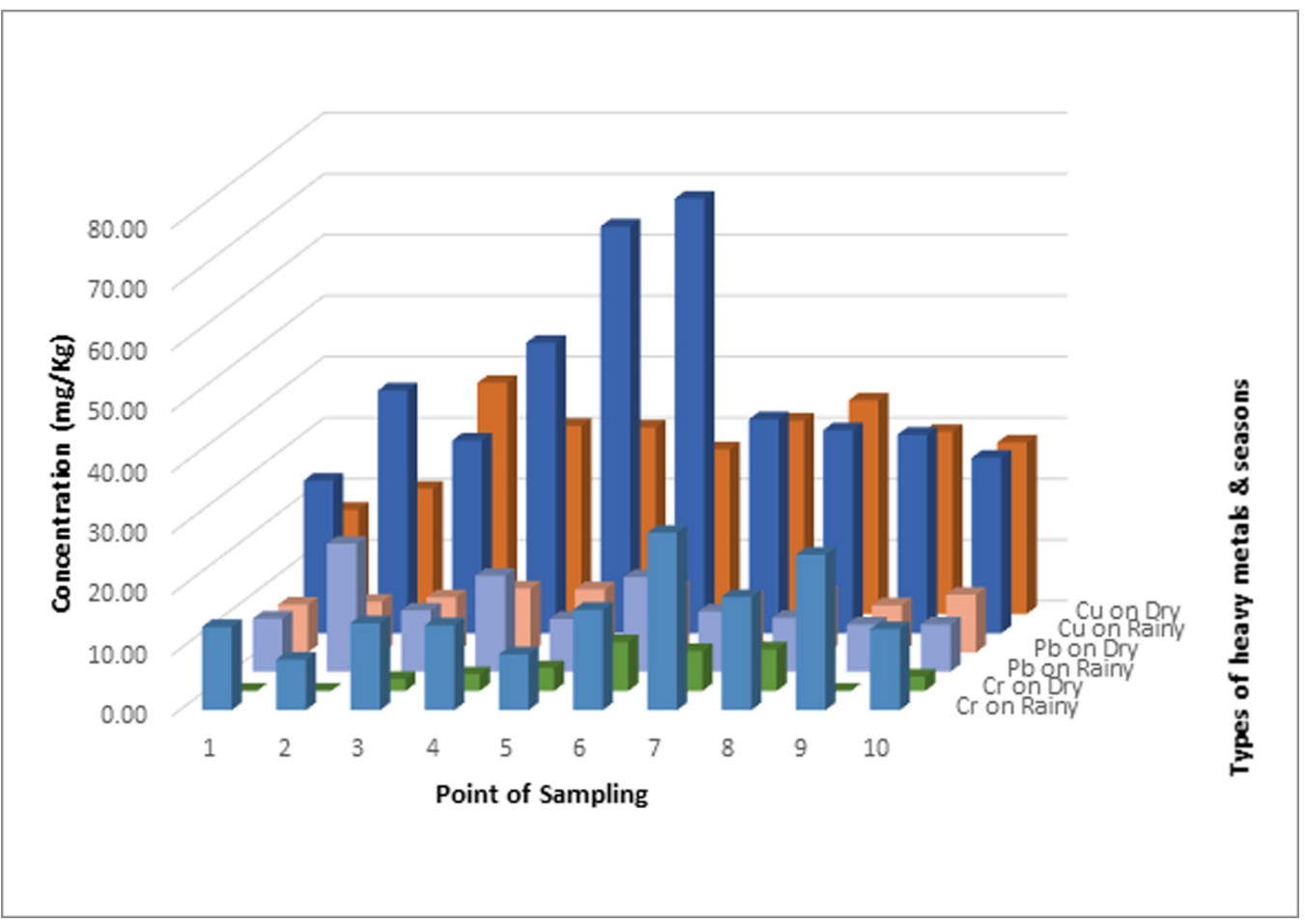

**Fig 3. Comparison of heavy metal concentrations (Cr, Cu, and Pb) in sediment compartments in the rainy and dry seasons.**

Determining the level of mobility of heavy metals that occur in the Cikijing River can be described by using the coefficient of organic-carbon ($K_{oc}$) and coefficient of octanol-water ($K_{ow}$) for each type of heavy metal analyzed. The calculation results of the heavy metal $K_{oc}$ and $K_{ow}$ values in the Cikijing River can be seen in the Tables 8 and 9.

Based on the results shown in Tables 8 and 9, it can be concluded that the source of heavy metal pollution at each point is affected by water discharge from land use around that point, which is dominated by industry, settlements, and agricultural land. This is supported by the Log $K_{oc}$ value Each type of heavy metal analyzed obtained a value (Log $K_{oc}$ > 3.6) which means that the heavy metal compound is difficult to move or has a very small percentage of mobility. This is supported by the results of the Log $K_{ow}$ value for each type of metal analyzed which is greater than zero, meaning that the heavy metal compounds Cr, Cu, Pb, and Zn have low solubility in water (hydrophobic).

## 3.3 Analysis of potential ecological risks based on heavy metal content in sediments

The concentration of heavy metals Cr, Cu, Pb, and Zn in the sediments of the Cikijing River will be used to assess potential ecological risks that can arise from the presence of these heavy metals in the environment. Natural concentrations of heavy metals can be interpreted as

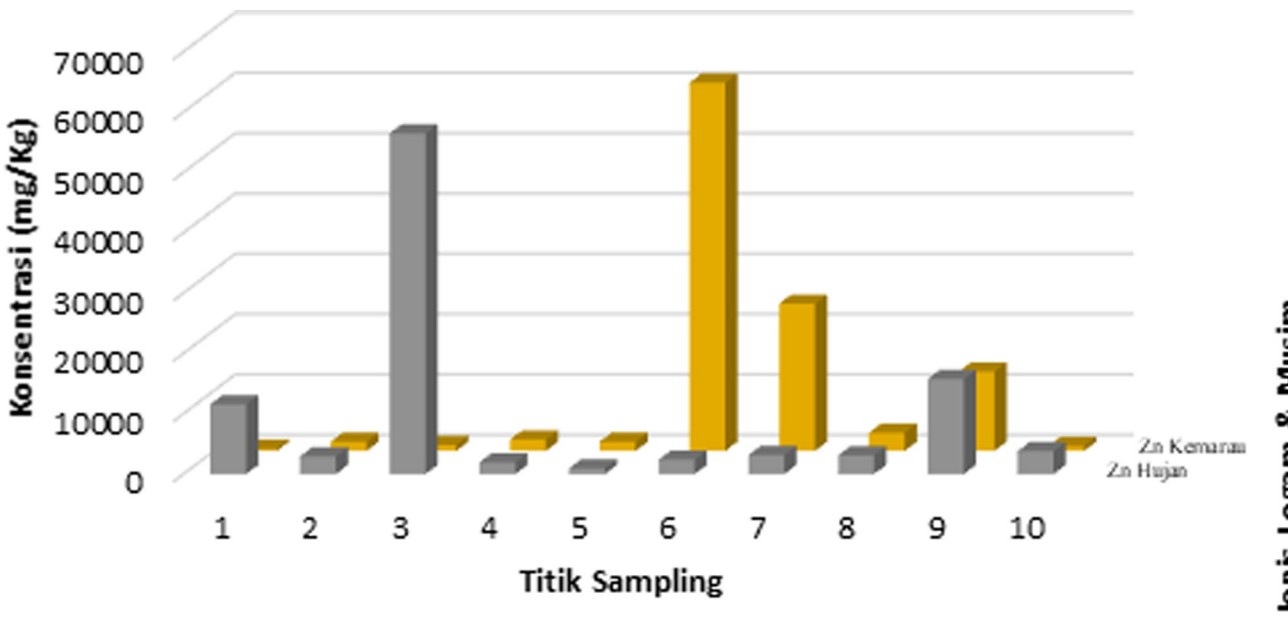

**Fig 4. Comparison of heavy metal concentrations (Zn) in sediment compartments in the rainy and dry seasons.**

concentration values obtained before industrial (pre-industrial) activities are carried out and pollute these waters [33]. Therefore, in this study, the quality standards used referred to the United States Environmental Protection Agency (USEPA) in 2000 titled "A Guidance Manual to Support the Assessment of Contaminated Sediments in Freshwater Ecosystems" and Turekian et. al (1961). This quality standard has also been used in the assessment of heavy metal pollution in the research of [34, 35]. The following is presented in Table 10 regarding the quality standards of heavy metals in sediments used in this study.

**3.3.1 Geoaccumulation index analysis of heavy metals in sediments.** The geoaccumulation index values ($I_{geo}$) of heavy metals Cr, Cu, Pb, and Zn in the sediments of the Cikijing River during the rainy and dry seasons are presented in Figs 5 and 6. The average value of the calculation of $I_{geo}$ in the sediments of the Cikijing River during the rainy and dry seasons for heavy metals Cr, Cu, and Pb have negative values ($I_{geo} < 0$) which means that the sediment quality is not polluted by chromium, copper, and lead. The results of calculating the average geoaccumulation index for Zn metal in the rainy season and dry season are in the range ($4 < I_{geo} < 5$). Meaning that the sediment quality of the Cikijing River ranges from heavily or severely polluted to very heavily polluted. The high concentration of Zn metal at these points could be due to the presence of sewerage channels for the textile industry prior to point 3 and several drainage channels for water from the community that entered the Cikijing River body and the use of chemical fertilizers containing the heavy metal Zn in community agricultural

**Table 8. Values of $K_{oc}$ and Log $K_{oc}$ of heavy metals Cr, Cu, Pb, and Zn in the Cikijing River.**

| | $K_{oc}$ & Log $K_{oc}$ | | | |
|---|---|---|---|---|
| | **Cr** | **Cu** | **Pb** | **Zn** |
| **Log $K_{oc}$** | 4.88 | 5.75 | 4.65 | 6,61 |
| **$K_{oc}$** | 75203 | 562288 | 44475 | 4052847 |
| **Category** | No Mobility | No Mobility | No Mobility | No Mobility |

**Table 9. Values of $K_{ow}$ and Log $K_{ow}$ of Heavy Metals Cr, Cu, Pb, and Zn in the Cikijing River.**

| | $K_{ow}$ & Log $K_{ow}$ | | | |
|---|---|---|---|---|
| | **Cr** | **Cu** | **Pb** | **Zn** |
| **Log $K_{ow}$** | 5.30 | 6.26 | 5.04 | 7,21 |
| **$K_{ow}$** | 197670 | 1834510 | 110488 | 16348067 |
| **Category** | Hydrophobic Compound | Hydrophobic Compound | Hydrophobic Compound | Hydrophobic Compound |

**Table 10. Reference quality standards for heavy metals in sediments used.**

| Heavy metal | Classification (mg/Kg) [36] | | | Natural concentration[37] |
|---|---|---|---|---|
| | **Class A** | **Class B** | **Class C** | |
| Cr | <43 | 43–110 | >110 | 90 |
| Cu | <32 | 32–150 | >150 | 39 |
| Pb | <36 | 36–130 | >130 | 23 |
| Zn | <120 | 120–460 | >460 | 120 |

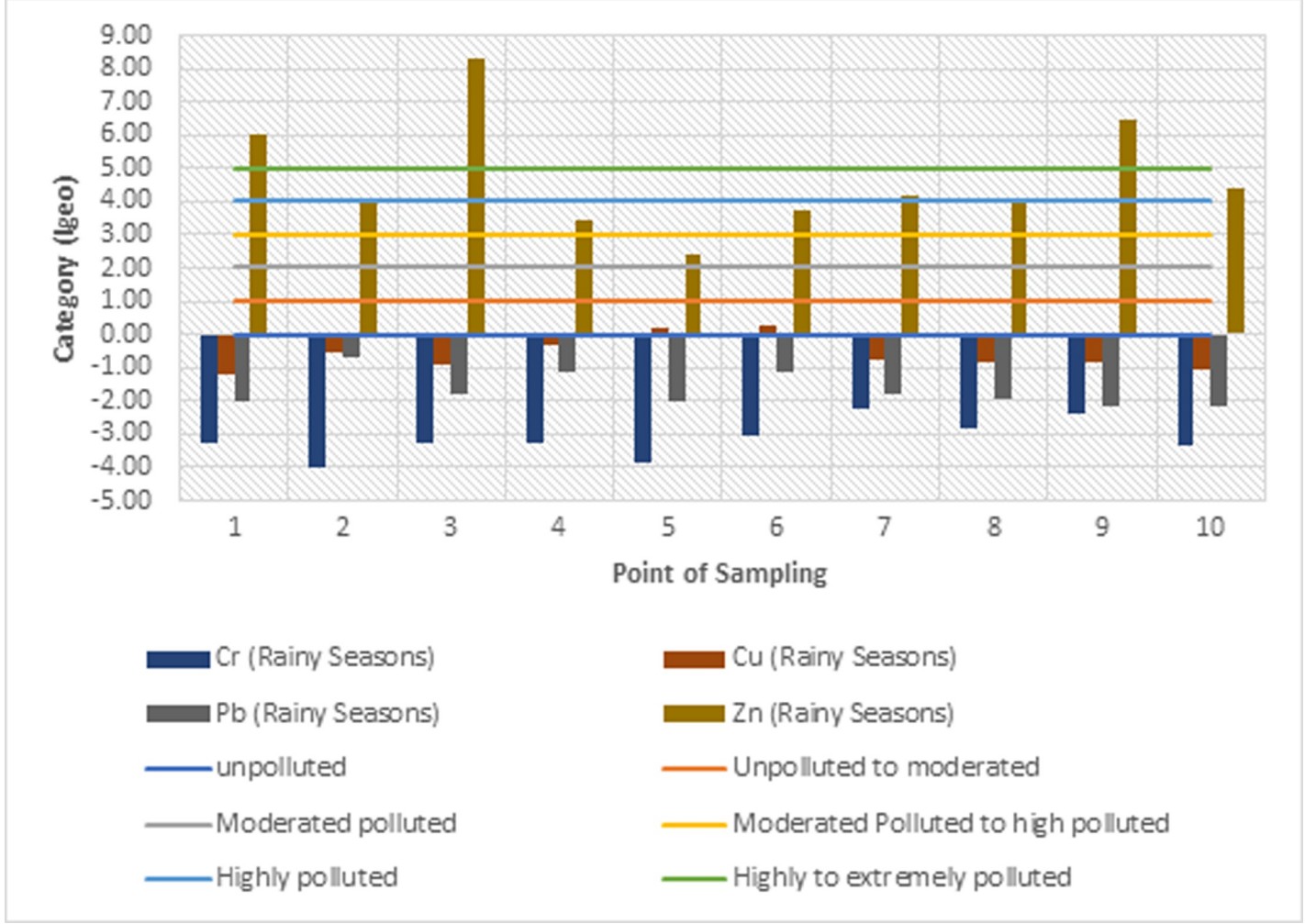

**Fig 5. Rainy season geoaccumulation index results.**

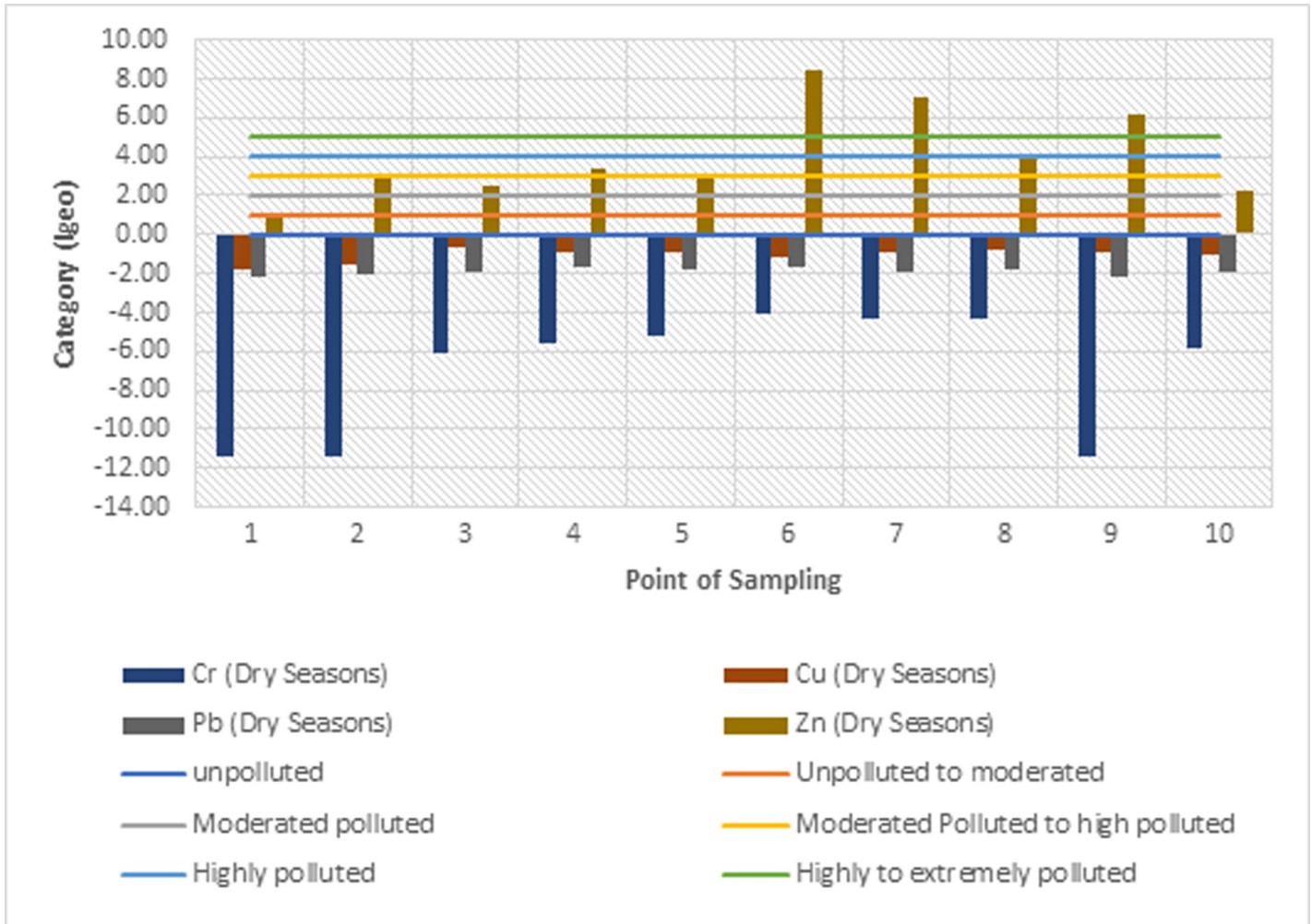

**Fig 6. Dry season geoaccumulation index results.**

activities. The high concentration of the heavy metal Zn can be sourced from residential waste, such as detergent waste [38].

**3.3.2 Analysis of pollutant load index of heavy metals in sediments.** The results of calculating the contaminant factor (CF) and pollution load index (PLI) values for Cr, Cu, Pb, and Zn both in the rainy and dry seasons are presented in Fig 7. The average value of the PLI from ten monitoring point in the Cikijing River during the rainy season and dry season were 1.37 and 0.73, respectively. This indicates that the sediment in the Cikijing River is polluted by several combinations of heavy metals only during the rainy season. The concentration of heavy metals during the rainy season is higher because it is influenced by erosion, rock erosion, and from the atmosphere that descends into river bodies along with rainwater [39].

**3.3.3 Analysis of potential ecological risk index of heavy metals in sediments.** Referring to Table 11, it is known that the potential ecological risks of the heavy metals Cr, Cu, and Pb from ten measurement points during the rainy and dry seasons have mild ecological risks. Meanwhile, the heavy metal Zn is the main element causing ecological risk in both seasons. Furthermore, the heavy metal potential toxicity response index (PERI) has a minimum value

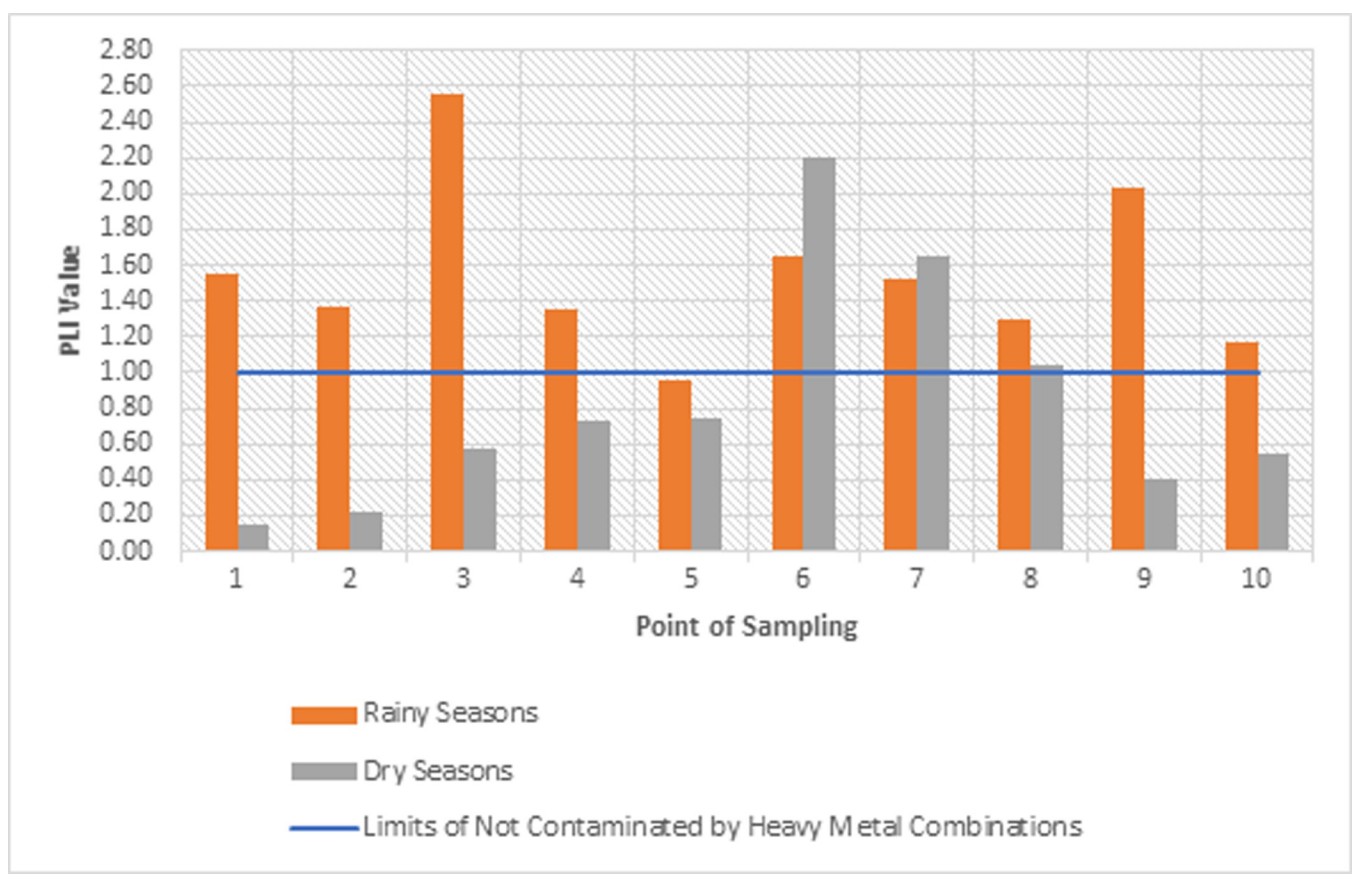

**Fig 7. Visualization of PLI results for the Cikijing River in the rainy and dry seasons.**

**Table 11. Results of the potential ecological risk index of heavy metals in the rainy and dry seasons.**

| Sampling Point | Ecological risk level of single-factor pollution (Rainy Season) | | | | PERI | Ecological risk level of single-factor pollution (Dry Season) | | | | PERI |
|---|---|---|---|---|---|---|---|---|---|---|
| | **Cr** | **Cu** | **Pb** | **Zn** | | **Cr** | **Cu** | **Pb** | **Zn** | |
| **1** | 0.635 | 3.91 | 1.21 | 96.5 | 102 | 0.002 | 2.67 | 1.09 | 3.08 | 6.84 |
| **2** | 0.386 | 6.22 | 2.91 | 24.5 | 34,0 | 0.002 | 3.21 | 1.15 | 12.6 | 17.0 |
| **3** | 0.663 | 4.94 | 1.39 | 471 | 478 | 0.091 | 5.92 | 1.26 | 8.15 | 15.4 |
| **4** | 0.647 | 7.44 | 2.18 | 16.2 | 26,5 | 0.128 | 4.82 | 1.46 | 15.6 | 22.0 |
| **5** | 0.426 | 10.4 | 1.20 | 7.98 | 20,0 | 0.174 | 4.78 | 1.44 | 12.8 | 19.2 |
| **6** | 0.765 | 11.1 | 2.15 | 20.1 | 34,2 | 0.372 | 4.22 | 1.46 | 509 | 515 |
| **7** | 1.35 | 5.48 | 1.37 | 26.4 | 34,6 | 0.305 | 4.95 | 1.22 | 203 | 210 |
| **8** | 0.863 | 5.20 | 1.22 | 26.0 | 33,3 | 0.316 | 5.47 | 1.35 | 25.0 | 32.1 |
| **9** | 1.19 | 5.09 | 1.08 | 132 | 139 | 0.002 | 4.67 | 1.07 | 109.4 | 115 |
| **10** | 0.619 | 4.50 | 1.08 | 31.7 | 37,9 | 0.109 | 4.39 | 1.32 | 6.87 | 12.7 |
| **Average** | 0.754 | 6.43 | 1.58 | 85.2 | 93.9 | 0.150 | 4.51 | 1.28 | 90.6 | 96.5 |
| **SD** | 0.308 | 2.49 | 0.620 | 141 | 140 | 0.138 | 0.971 | 0.147 | 160 | 160 |
| **CV(%)** | 40,8 | 38,7 | 39,2 | 165 | 149 | 92,0 | 21,5 | 11,5 | 177 | 166 |
| **Category** | Mild Ecological Risk | Mild Ecological Risk | Mild Ecological Risk | High Ecological Risk | Mild Ecological Risk | Mild Ecological Risk | Mild Ecological Risk | Mild Ecological Risk | High Ecological Risk | Mild Ecological Risk |

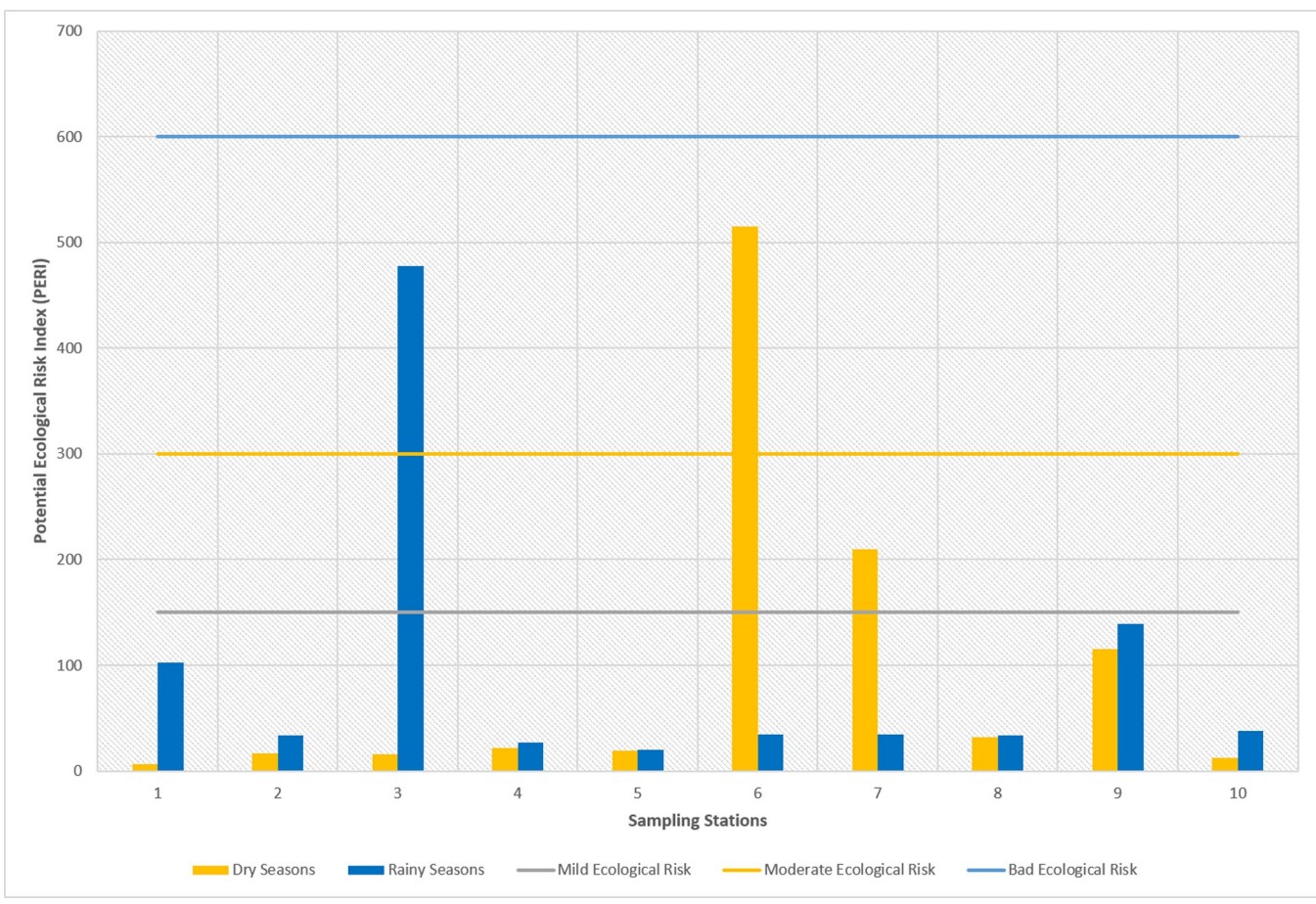

**Fig 8. Visualization of potential ecological risks in the Cikijing River in the rainy and dry seasons.**

of 6.84 at sampling point 1, which is in the mild ecological risk category (PERI <150), while the maximum value is 514.74 at point 6, which is in the bad ecological risk category (300 < PERI < 600). The PERI average values for 10 measurement points both in the rainy and dry seasons were 93.94 and 96.49, respectively, which indicates that the sediment in the Cikijing River has a mild ecological risk. The PERI of the Cikijing River during the rainy and dry seasons are presented in Fig 8.

**3.3.4 Matrix analysis of differences in sediment rating indices used (Igeo, PLI, and PERI).** The analysis matrix regarding the differences in sediment assessment indices used, namely the geoaccumulation index ($I_{geo}$), pollutant load index (PLI), and Ecological Potential Risk Index (PERI) is presented in Table 12.

## 3.4. Correlation statistical analysis

**3.4.1 Analysis of physical-chemical parameter relationship to heavy metal concentration in water compartment.** Based on the results of the normality test presented in Table 13, three data are normally distributed both (independent variables) and (dependent variables), namely Pb, TSS, and DO concentration data in the dry season, so it is necessary to carry out a homogeneity test using one-way ANOVA to ensure that the data is suitable for use. Parametric analysis test using Pearson correlation type.

**Table 12. Matrix analysis of differences in geoaccumulation index, pollutant load index, and ecological risk potential index.**

| Parameter | | Geoaccumulation Index (*Igeo*) | Pollutant Load Index (PLI) | Ecological Potential Risk Index (PERI) |
|---|---|---|---|---|
| **Definition and Advantages** | | To determine the level of heavy metal pollution in sediments by considering the effects of variations in the layers of the earth [33]. | To determine the status of metal pollution in sediments in a waters, whether the status is not polluted or polluted by a collection of metals [35]. | That represents the sensitivity of the biological community to all toxic/toxic substances present in sediments. The Potential Ecological Risk Index can effectively reflect the comprehensive potential effects of several metal contaminants in sediments on the ecological environment by increasing the toxicity response coefficient based on the metal content [40]. |
| **Disadvantages** | | Given the inhomogeneous geological conditions of the sediments in many cases, geochemical background information is often represented in terms of uncertain intervals rather than concrete values and current geoaccumulation indices cannot deal with uncertainty like that. Sediments often contain many heavy metal species [41]. | Although the Pollutant Load Index can determine whether a sediment in waters is polluted or not by several combinations of heavy metals, the Pollutant Load Index has so far not been able to categorize the level of pollution and cannot show the comprehensive impact of several metal elements on the environment. | The Potential Ecological Risk Index (PERI) in general can represent the level of ecological risk posed by heavy metal pollution in the sediment compartment in the waters, but to get representative and comprehensive results it is necessary to measure more types of heavy metals because the calculation of the PERI value is the sum or accumulation the total value of the level of ecological risk for each type of heavy metal analyzed. |
| **Previous research** | | Metals (Al, Cr, Co, Cu, Fe, Hg, Mn, Pb, and Zn) in Doce River Sediments, Brazil [42]. | Metals (Cr, Cu, Fe, Mn, Ni, and Zn) in Segara Anakan, Cilacap [43]. | Metals (Cd, Cr, Cu, Ni, Pb, and Zn) in Luan River Sediments, North China [44]. |
| | | Metals (Cd, Co, Cu, Pb, and Zn) in Ghalechay River Sediments, Iran [45]. | Metals (Fe, Mn, Zn, Cu, Pb, Ni, Co, Cd, Cr, and Hg) in coastal sediments of the Gulf of Suez, Egypt [46]. | Metals (Al, As, Cd, Cr, Cu, Fe, Mn, Pb, and Zn) in Lancang River Sediments, China [47]. |
| **Current Research** | Cr | Unpolluted | Cikijing River is polluted by heavy metals (PLI>1) | low (mild) potential ecological risk (PERI was found 93.94 and 96.49) |
| | Cu | Unpolluted | | |
| | Pb | Unpolluted | | |
| | Zn | heavily polluted | | |
| **Previous Researches** | Cr | Moderately to heavily polluted [38] | Benin River is unpolluted by heavy metal (PLI<1) [48] | low potential ecological risk (PERI was found 90.91) [49] |
| | Cu | Unpolluted [38] | | |
| | Pb | heavily polluted [38] | | |
| | Zn | heavily polluted [38] | | |

Based on the normality and homogeneity test results using the oneway ANOVA test presented in Table 14, the type of correlation analysis used for both rainy and dry season data uses Spearman's correlation (non-parametric). The relationship between heavy metal concentrations and physical-chemical parameters during the rainy season shows that there is a significant relationship between the heavy metal Pb and physical-chemical parameters such as TDS, salinity, and electrical conductivity (sig. <0.05). The relationship between the concentration of the heavy metal zinc (Zn) and the physical-chemical parameters of the water compartment, which has a strong correlation with TSS (0.606) and DO (0.543) and is quite strong for the temperature parameter (0.407), but the three correlations are not significant (sig. >0.05). As with the heavy metal Pb, the relationship between pH and the concentration of the heavy metal Zn in the water compartment.

**Table 13. Homogeneity test of Pb, TSS, and DO concentration data in the dry season.**

| Test of Homogeneity of Variances | | | | |
|---|---|---|---|---|
| | **Levene Statistics** | **df1** | **df2** | **Sig.** |
| DO (mg/L) | 0.235 | 1 | 18 | 0.633 |
| TSS (mg/L) | 3.95 | 1 | 18 | 0.062 |
| Pb (mg/L) | 6.17 | 1 | 18 | 0.023 |

**Table 14. Correlation test results of physical-chemical parameters on heavy metal concentrations in the rainy season.**

| | | | TDS (mg/L) | TSS (mg/L) | pH | DO (mg/L) | Salinity (ppt) | DHL(uS/cm) | Temperature (oC) |
|---|---|---|---|---|---|---|---|---|---|
| | | | **Correlations** | | | | | | |
| Spearman's rho | Pb (mg/L) (Rain) | Correlation Coefficient | 0.936** | 0.125 | -0.024 | 0.485 | 0.938** | 0.936** | 0.462 |
| | | Sig. (2-tailed) | 0,000 | 0.730 | 0.947 | 0.156 | 0,000 | 0,000 | 0.179 |
| | | N | 10 | 10 | 10 | 10 | 10 | 10 | 10 |
| | Zn (mg/L) (Rain) | Correlation Coefficient | 0.109 | 0.606 | -0.018 | 0.543 | 0.111 | 0.109 | 0.407 |
| | | Sig. (2-tailed) | 0.763 | 0.064 | 0.960 | 0.105 | 0.760 | 0.763 | 0.243 |
| | | N | 10 | 10 | 10 | 10 | 10 | 10 | 10 |

**. Correlation is significant at the 0.01 level (2-tailed); *. Correlation is significant at the 0.05 level (2-tailed).

The results of the Spearman-type correlation between physical-chemical parameters on heavy metal concentrations during the dry season presented in Table 15, indicate that the heavy metal Pb has a strong and directly proportional (positive) correlation with TDS, pH, salinity, electrical conductivity, and temperature (0.523; 0.522 ; 0.522; 0.561; and 0.587) but not significant (sig > 0.05). While the correlation relationship between physical-chemical parameters for the heavy metal Zn in the dry season; there is a strong and unidirectional (positive) correlation between TDS, pH, DO, salinity and electrical conductivity (0.673; 0.591; 0.709; 0.755; and 0.657) as well as significant (sig. <0.05). The TSS and temperature parameters had a weak and positive relationship with the concentration of heavy metal Zn (0.203 and 0.165) but were not significant (sig. > 0.05).

**3.4.2 Analysis of relationship between sediment texture and organic matter with heavy metal concentrations in sediment compartments.** The homogeneity test presented in Tables 16 and 17 was carried out using one-way ANOVA for parameters that had normally distributed data to ensure that the data was suitable for parametric analysis testing using the Pearson correlation type.

Based on the results of correlation tests for both Pearson correlation and Spearman correlation, results showed in Table 18 that there was no significant relationship between heavy metal concentrations (Cr, Cu, Pb, and Zn) and total organic matter in sediments both in the rainy and dry seasons (sig. > 0.05). Similar results were obtained in the study by Helali et al. (2013) [50], namely that there was no significant correlation between the heavy metals Pb and Cr in the Mejerda River, Tunisia. Different results were also found in Maslukah's research (2013) [51], which stated that organic matter is the most important geochemical component in controlling the binding of heavy metals from sediments in waters. However, there is no significant correlation between organic matter and heavy metal concentrations Cr, Cu, Pb, and Zn shown in Tables 19–21 that

**Table 15. Correlation test results of physical-chemical parameters on heavy metal concentrations in the dry season.**

| | | | TDS (mg/L) | TSS (mg/L) | pH | DO (mg/L) | Salinity (ppt) | DHL(uS/cm) | Temperature (oC) |
|---|---|---|---|---|---|---|---|---|---|
| | | | **Correlations** | | | | | | |
| Spearman's rho | Pb (mg/L) (Dry) | Correlation Coefficient | 0.523 | 0.077 | 0.522 | 0.243 | 0.522 | 0.561 | 0.587 |
| | | Sig. (2-tailed) | 0.121 | ,832 | 0.122 | 0.498 | 0.121 | 0.092 | 0.074 |
| | | N | 10 | 10 | 10 | 10 | 10 | 10 | 10 |
| | Zn (mg/L) (Dry) | Correlation Coefficient | 0.673* | 0.203 | 0.591 | 0.709* | 0.755* | 0.657* | 0.165 |
| | | Sig. (2-tailed) | 0.033 | 0.574 | 0.072 | 0.062 | 0.012 | 0.039 | 0.649 |
| | | N | 10 | 10 | 10 | 10 | 10 | 10 | 10 |

**. Correlation is significant at the 0.01 level (2-tailed); *. Correlation is significant at the 0.05 level (2-tailed).

**Table 16. Homogeneity test of Cr and total organic matter data in sediment during the rainy season.**

| Test of Homogeneity of Variances | | | | |
|---|---|---|---|---|
| | Levene Statistics | df1 | df2 | Sig. |
| Cr (mg/Kg) | 3.34 | 1 | 18 | 0.084 |
| Total Organic Matter (%) | 0.007 | 1 | 18 | 0.936 |

**Table 17. Homogeneity test of heavy metal concentration data (Cr, Cu, and Pb), total organic matter, and sediment texture in the dry season.**

| Test of Homogeneity of Variances | | | | |
|---|---|---|---|---|
| | Levene Statistics | df1 | df2 | Sig. |
| Cr (mg/Kg) | 3.34 | 1 | 18 | 0.084 |
| Cu (mg/Kg) | 6.34 | 1 | 18 | 0.022 |
| Pb (mg/Kg) | 13.1 | 1 | 18 | 0.002 |
| Total Organic Matter (%) | 0.01 | 1 | 18 | 0.936 |
| Sand (%) | 0.98 | 1 | 18 | 0.335 |
| Silt & Clay (%) | 0.98 | 1 | 18 | 0.335 |

**Table 18. Pearson correlation test results of the relationship between heavy metal Cr concentration and total organic matter in the rainy season.**

| Correlations | | |
|---|---|---|
| | | Total Organic Matter (%) (Rain) |
| Cr (mg/Kg) | Pearson Correlation | 0.333 |
| | Sig. (2-tailed) | 0.348 |
| | N | 10 |

**Table 19. Spearman correlation test results of the relationship between heavy metal concentrations (Cr, Cu, Pb, Zn) on total organic matter and sediment texture in the rainy season.**

| Correlations | | | | | |
|---|---|---|---|---|---|
| | | | Total Organic Matter (%) | Sand (%) | Silt & Clay (%) |
| Spearman's rho | Cr (mg/Kg) | Correlation Coefficient | - | -0.067 | 0.067 |
| | | Sig. (2-tailed) | - | 0.855 | 0.855 |
| | | N | | 10 | 10 |
| | Cu (mg/Kg) | Correlation Coefficient | 0.285 | -0.309 | 0.309 |
| | | Sig. (2-tailed) | 0.425 | 0.385 | 0.385 |
| | | N | 10 | 10 | 10 |
| | Pb (mg/Kg) | Correlation Coefficient | 0.292 | 0.018 | -0.018 |
| | | Sig. (2-tailed) | 0.413 | 0.960 | 0.960 |
| | | N | 10 | 10 | 10 |
| | Zn (mg/Kg) | Correlation Coefficient | -0.224 | 0.297 | -0.297 |
| | | Sig. (2-tailed) | 0.533 | 0.405 | 0.405 |
| | | N | 10 | 10 | 10 |

**. Correlation is significant at the 0.01 level (2-tailed).

**Table 20. Pearson correlation test results of the relationship between heavy metal Cr concentrations on total organic matter and sediment texture in the dry season.**

| | | **Correlations** | | |
|---|---|---|---|---|
| | | **Total Organic Matter (%)** | **Sand (%)** | **Silt & Clay (%)** |
| Cr (mg/Kg) | Pearson Correlation | -0.194 | -0.721* | 0.721* |
| | Sig. (2-tailed) | 0.591 | 0.019 | 0.019 |
| | N | 10 | 10 | 10 |

*. Correlation is significant at the 0.05 level (2-tailed).

presumably because these metals in the sediment bind to different fractions other than the organic matter fraction. Based on other studies, there are several types of fractions besides the organic matter fraction, namely the exchangeable fraction, carbonate fraction, reducible fraction (Fe-Mn Oxide), and residual fraction [45, 52] . The following in Table 22 are the results of measurements of sediment texture at sampling locations both during the rainy and dry seasons along with the classification of sediment textures based on references from Cornell University [53].

Based on the results of the Pearson correlation test presented in Table 20 on a review of the correlation between sediment texture and heavy metal Cr in the dry season, it shows that there is a significant correlation (sig. <0.05) between the percentage of sand and heavy metal Cr, which has a strong negative correlation (-0.721) and has strong positive correlation (0.721) for the relationship between the texture of silt and clay sediments and the concentration of the heavy metal Cr. The results of this study are consistent with the results of previous studies, namely that sediment texture has a significant correlation with Cr metal with the conclusion that finer sediment textures (silt and clay) will bind more heavy metals when compared to coarser sediment textures (sand). The percentage of sediments with a high silt tends to be high in metal content.

Meanwhile, based on the results of other correlation tests shown in Tables 19–21, there was no significant correlation between the texture of sand and mud (silt and clay) sediments and the heavy metals Cr in the rainy season and Pb, Cu, and Zn both in the rainy and dry seasons. The results of this study are similar to the research conducted by Muflih in 2014 in the coastal area of Tangerang, which found that there was no relationship between sediment texture and heavy metal concentrations of Cu and Pb. Other studies obtained insignificant correlation results (sig. > 0.05) for the relationship of sediment texture (sand, silt, and clay) to heavy metals Cu, Pb, and Zn in the Ghalechay River, Iran [50].

**Table 21. Spearman correlation test results of the relationship between heavy metal concentrations (Cu, Pb, Zn) on total organic matter and sediment texture in the dry season.**

| | | | **Correlations** | | |
|---|---|---|---|---|---|
| | | | **Total Organic Matter (%)** | **Sand (%)** | **Silt & Clay (%)** |
| Spearman's rho | Cu (mg/Kg) | Correlation Coefficient | 0.042 | -0.018 | 0.018 |
| | | Sig. (2-tailed) | 0.907 | 0.960 | 0.960 |
| | | N | 10 | 10 | 10 |
| | Pb (mg/Kg) | Correlation Coefficient | -0.406 | -0.430 | 0.430 |
| | | Sig. (2-tailed) | 0.244 | 0.214 | 0.214 |
| | | N | 10 | 10 | 10 |
| | Zn (mg/Kg) | Correlation Coefficient | 0.176 | -0.091 | 0.091 |
| | | Sig. (2-tailed) | 0.627 | 0.803 | 0.803 |
| | | N | 10 | 10 | 10 |

**Table 22. The results of measurements of sediment texture at sampling locations both during the rainy and dry seasons.**

| Stations of Sampling | Seasons | | | | Sediment Texture Classes |
|---|---|---|---|---|---|
| | Rainy | | Dry | | |
| | Sand (%) | Mud(Silt & Clay) (%) | Sand (%) | Mud (Silt & Clay) (%) | |
| 1 | 97,26 | 2.74 | 93.96 | 6.04 | Sand particles range in size from 0.05–2.0 mm, Silt ranges from 0.002–0.05 mm |
| 2 | 97.64 | 2.36 | 84,68 | 15,32 | clay fraction is made up of particles less than 0.002 mm |
| 3 | 94.96 | 5.04 | 91.68 | 8,32 | |
| 4 | 84,24 | 15.76 | 91.20 | 8.80 | |
| 5 | 93,82 | 6,18 | 93.36 | 6,64 | |
| 6 | 96.35 | 3.65 | 97.88 | 2,12 | |
| 7 | 94,26 | 5,74 | 95.57 | 4,43 | |
| 8 | 97.40 | 2.60 | 97.42 | 2.58 | |
| 9 | 96.54 | 3.46 | 85.92 | 14.08 | |
| 10 | 95.72 | 4,28 | 98.21 | 1.79 | |

Based on the research results, a correlation has been obtained between the concentration of dissolved heavy metals and the physical-chemical parameters considered in Table 23. Meanwhile, the correlation between the concentration of heavy metals in the sediment compartment with the organic matters parameters and sediment size has been shown in Table 24.

This study have found that there are contamination of heavy metals in Cikijing river, still there is a limitation of this study. Sampling activities in this study were carried out in months which are representative of the wet season and dry season. It would be better if the sampling was carried out every month (January to December) so the results can show the characteristics of heavy metal contamination at all conditions.

Based on the results of this study, there are several recommendations that should be taken by stakeholders, especially for those who giving business permissions to industries around the Cikijing River. They have to strictly monitor the performance of each wastewater treatment plant and force each industry to carry out conservation efforts around the Cikijing River.

## 4 Conclusion

Based on the research results, it was found that the water quality of the Cikijing River based on heavy metals Cr, Cu, Pb, and Zn with an assessment using the pollution index was classified as a river not polluted by heavy metals.

The potential ecological risk caused by heavy metal contamination of Cr, Cu, Pb, and Zn in the sediments of the Cikijing River is classified as a mild potential ecological risk. This is supported by the assessment of the geoaccumulation index in the Cikijing River which is categorized as unpolluted for the heavy metals Cr, Cu, and Pb ($I_{geo} < 0$), and heavily polluted to very heavily polluted for the heavy metal Zn ($4 < I_{geo} \leq 5$). The Pollution Load Index is useful for determining the pollution status of several heavy metal contaminants in sediments, and it is found that the Cikijing River is classified as heavily polluted by several heavy metals (PLI > 1).

Based on the results of the correlation test that has been carried out, it was found that the physical-chemical parameters of total dissolved solids, salinity, and electrical conductivity have a strong and significant correlation or relationship to the concentration of heavy metals Pb and Zn in Cikijing River water. While in the sediment compartment, it is known that there is no significant relationship between the concentration of heavy metals (Cr, Cu, Pb, and Zn) with total organic matter in sediments both in the rainy and dry seasons. Sediment texture was found to have a strong and significant correlation with heavy metal Cr concentrations, i.e.

**Table 23. Relationship between dissolved heavy metal concentrations and physical-chemical parameters.**

| | | Heavy Metals | | | | Literature |
|---|---|---|---|---|---|---|
| | | Cr | Cu | PB | Zn | |
| physicochemical Parameters | Temperature | concentration data is below the detection limit of the AAS tool | concentration data is below the detection limit of the AAS tool | The correlation is quite strong and positive | Weak and Positive Correlation | According to Li et al., (2013) [54] that high temperatures (30–35˚C) in surface water can increase the rate of release of heavy metals from complex compounds (desorption) compared to lower temperatures. |
| | DO | | | The correlation is quite strong and positive | Weak and Positive Correlation | DO does not have a significant effect on dissolved heavy metals (Huang et al., 2017) [55] |
| | TDS | | | Very Strong and Positive Correlation | Strong and Positive Correlation | The increase in the TDS value occurs because the current speed increases so that organic matter and colloids in the form of chemical compounds dissolved in water tend to increase (Effendi, 2003) [56] |
| | TSS | | | The correlation is quite strong and positive | Weak and Positive Correlation | High TSS tends to increase the concentration of dissolved heavy metals due to the desorption process of heavy metals from the sediment compartment |
| | Salinity | | | Very Strong and Positive Correlation | Strong and Positive Correlation | The decrease in salinity due to the desalination process will cause an increase in the toxic power of heavy metals and greater levels of heavy metal bioaccumulation (Erlangga, 2007) [57] |
| | EC | | | Very Strong and Positive Correlation | Strong and Positive Correlation | The EC value can accelerate and increase the heavy metal desorption process which results in an increase in heavy metals in water (Alsaffar et al., 2016) [58] |
| | pH | | | Weak and Negative Correlation | Weak and Negative Correlation | An increase in the concentration of dissolved heavy metals coincides with a decrease in pH because metal ions are released from complex compounds and become free ions in water bodies. Liu et al., (2013) [59] |

finer sediment textures (silt and clay) will bind more heavy metal Cr when compared to coarser sediment textures (sand).

**Table 24. Relationship between heavy metal concentrations in sediment compartments and physical-chemical parameters.**

| | | Heavy Metals | | | | Literature |
|---|---|---|---|---|---|---|
| | | Cr | Cu | PB | Zn | |
| physicochemical Parameters | Total Organic Matters | there is no significant relationship | there is no significant relationship | there is no significant relationship | there is no significant relationship | Thomas and Bendell-Young (1998) [60] in Maslukah (2013) [51], who stated that organic material is the most important geochemical component in controlling the binding of heavy metals from sediments in waters. |
| | Sand | there is no significant correlation | there is no significant correlation | there is no significant correlation | there is no significant correlation | Ghazban et al. (2015) [45] that the correlation results are not significant for the relationship between sediment texture (sand, silt, and clay) to the heavy metals Cu, Pb, and Zn in the Ghalechay River, Iran. The metal content will increase with the finer grain size of the sediment, the finer the sediment texture. having a solid form makes it easier to bind metals in the deposition process (Maslukah, 2013) [51]. |
| | Silt & Clay | there is no significant correlation | there is no significant correlation | there is no significant correlation | there is no significant correlation | |

## Supporting information

**S1 File.**
(DOCX)

## Author Contributions

**Conceptualization:** M. Wijaya.

**Data curation:** Mariana Marselina, M. Wijaya.

**Formal analysis:** M. Wijaya.

**Funding acquisition:** Mariana Marselina.

**Investigation:** M. Wijaya.

**Methodology:** M. Wijaya.

**Supervision:** Mariana Marselina.

**Validation:** Mariana Marselina.

**Writing – original draft:** M. Wijaya.

**Writing – review & editing:** Mariana Marselina, M. Wijaya.

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
