## [Decision Letter · Decision Letter 0]

17 May 2023

PONE-D-23-01059Analysis of Water Quality and Potential Ecological Risk Based on Heavy Metal Content (Cr, Cu, Pb, and Zn) in the Cikijing River, Rancaekek District, West Java, IndonesiaPLOS ONE

Dear Dr. Marselina,

Thank you for submitting your manuscript to PLOS ONE. After careful consideration, we feel that it has merit but does not fully meet PLOS ONE’s publication criteria as it currently stands. Therefore, we invite you to submit a revised version of the manuscript that addresses the points raised during the review process.

We look forward to receiving your revised manuscript.

Kind regards,

Xiaoshan Zhu, Ph.D.

Academic Editor

PLOS ONE

3. We note that [Figure 1] in your submission contain [map/satellite] images which may be copyrighted. All PLOS content is published under the Creative Commons Attribution License (CC BY 4.0), which means that the manuscript, images, and Supporting Information files will be freely available online, and any third party is permitted to access, download, copy, distribute, and use these materials in any way, even commercially, with proper attribution. For these reasons, we cannot publish previously copyrighted maps or satellite images created using proprietary data, such as Google software (Google Maps, Street View, and Earth). For more information, see our copyright guidelines: http://journals.plos.org/plosone/s/licenses-and-copyright.

Natural Earth (public domain): http://www.naturalearthdata.com/.

Reviewers' comments:

Reviewer's Responses to Questions

**Comments to the Author**

1. Is the manuscript technically sound, and do the data support the conclusions?

Reviewer #1: Yes

Reviewer #2: No

2. Has the statistical analysis been performed appropriately and rigorously? 

Reviewer #1: Yes

Reviewer #2: No

3. Have the authors made all data underlying the findings in their manuscript fully available?

Reviewer #1: Yes

Reviewer #2: Yes

4. Is the manuscript presented in an intelligible fashion and written in standard English?

Reviewer #1: No

Reviewer #2: No

5. Review Comments to the Author

Reviewer #1: Please refer to the attached file for my report. The subject addressed is within the scope of the journal.

However, the manuscript, in its present form, contains several weaknesses. Appropriate revisions to the mentioned points should be undertaken in order to justify recommendation for publication.

Reviewer #2: The manuscript “Analysis of Water Quality and Potential Ecological Risk Based on Heavy Metal Content (Cr, Cu, Pb, and Zn) in the Cikijing River, Rancaekek District, West Java” is a manuscript assessed heavy metals in West Java. However, I recommend the following suggestions before the article can be accepted for publication I encourage potential justification/revision of the comments/suggestions during its revision:

*Author must be change the title of this paper and need to include components like water and sediment.

*Revise the highlighted points with specific and focused results. The existing highlights are simply stated without pinpointing the novel study findings.

*The abstract should stand alone with focused results. Thus, the authors are advised to update the abstract and improve it with specific data and other relevant results from this current research.

#An updated version of abstract is incorporated in the revised manuscript.

*Minor typos in English and syntax must be thoroughly improved in the manuscript. Please make double-check during revision.

Abstract:

Please rewrite the abstract

There is no statistical analysis such as Principle component analysis and cluster analysis. In addition, authors also mention or give suggestion for the management framework to reduce HMs.

1.Introduction

Author must rewrite this section

Why HMs harmful environment?

What is the significant of this studies?

Describe Literature (HMs) in global as well as authors country

How it is related in food chain?

Author can follow these papers:

a. https://www.tandfonline.com/doi/full/10.1080/15569543.2021.1891936

b. https://doi.org/10.1016/j.scitotenv.2020.144637

c. https://link.springer.com/article/10.1007/s11356-021-15353-9

d. https://doi.org/10.1016/j.ijsrc.2021.09.002

e. https://www.tandfonline.com/doi/full/10.1080/15320383.2021.1923644

f. https://link.springer.com/article/10.1007/s11356-022-22122-9

g. https://link.springer.com/article/10.1007/s11356-021-17153-7

2.Method and materials

1. Study area must be described significantly why authors select this area.

2. Authors include the station id in the map like S1, S2…….

3.Author can describe all pollution in single table to reduce the paper length and summarize it.

Please this one- https://www.tandfonline.com/doi/full/10.1080/15320383.2021.1923644

3.Results and Discussion

1. In this section, all of the figures are totally blur and must be redraw.

2. PCA and cluster analysis must be included for source identification

3. Comparison table must be needed to compare their studies with others studies in the globe.

4. Conclusion

I think author should give advice for government and stakeholders and so on.

6. PLOS authors have the option to publish the peer review history of their article (what does this mean?). If published, this will include your full peer review and any attached files.

Reviewer #1: **Yes: **Yalçın TEPE

Reviewer #2: No

---

## [Author Response · Author response to Decision Letter 0]

30 Jul 2023

Reviewer 1 Comments or Revision

1 The limitations of this study, suggested improvements of this work should be highlighted  Limitation of this study has been added in the manuscript

2 Manuscript could be substantially improved by relying and citing more on recent literatures about contemporary real-life case studies on sustainability and/or metal accumulations such as the followings:

https://doi.org/10.1016/j.iswcr.2018.09.001

https://doi.org/10.1080/15275922.2020.1728433

https://doi.org/10.1007/s10661-022-10490-1  Those papers have been discussed and cited in Introduction section especially at paragraph 3 and 4

3 The authors should link the concentration of the heavy metals in the study area to the geology Have been added in Introduction (paragraph 8)

4 It would be appropriate if a comparison table was made with the values found around the world  Have been added in Table 12

5 Please check the legends on the figures and/or tables for completeness  The legend in the figures and/or tables has been corrected and added

6 Two sampling times were set up as rainy and dry seasons in 2022. A full explanation have been added in the materials and methods section

7 Specify the temperature and time period for the drying process. A full explanation has been added in the Data Collection section

8 Did you use grinder and sieve prior to digestion of sediment samples? We use sieve prior identification of sediment texture

9 The recovery results of the certified reference material in a separate table to show your % more clearly  The results of measurements of sediment texture at sampling locations both during the rainy and dry seasons has been added in Table 13

10 The interaction between shale, sediment, and heavy metal should be re-explained in detail and given correctly  Thank you very much for the suggestion, the interaction between heavy metals and shale, sediment is based on field data and processed using the IBM SPSS Statistics application

11 The authors should also compare the results with the standard acceptable levels established by the Minister of Environment Number 115 of 2003  Have been added in Results and Discussion (Paragraph 2)

12 Improvements are required for the tables and figs  Thanks for the suggestion, the figures and tables have been fixed

Reviewer 2 Comments or Revision

1 Author must be change the title of this paper and need to include components like water and sediment  The title of this paper has been revised with include components of water and sediment 

The title become:

“Heavy Metals in Water and Sediment of Cikijing River, Rancaekek District, West Java: Contamination Distribution and Ecological Risk Assessment”

2 Author must rewrite this section

Why HMs harmful environment?  Narration about HMs have been added in Introduction (paragraph 2)

Describe Literature (HMs) in global as well as authors country  Have been added in Introduction (paragraph 6)

How it is related in food chain?  Have been added in Introduction (paragraph 5)

What is the significant of this studies?  Have been added in Introduction (Last paragraph)

3 Author can follow these papers:

a. https://www.tandfonline.com/doi/full/10.1080/15569543.2021.1891936

b. https://doi.org/10.1016/j.scitotenv.2020.144637

c. https://link.springer.com/article/10.1007/s11356-021-15353-9

d. https://doi.org/10.1016/j.ijsrc.2021.09.002  Those papers have been discussed and cited in Introduction section especially at paragraph 5

4 PCA and cluster analysis must be included for source identification (MM) Actually, in this study, the location itself which is Cikijing River is specific location or area that consist of several industries, especially textile industry. So we consider, source identification not really have to be done, because this location is a specific location with specific type of pollution (industries pollutants)

5 I think author should give advice for government and stakeholders and so on.  Advice for government have been added into the manuscript

6 The abstract should stand alone with focused results.  Have been added in Abstract (Sentences 4-7)

7 Study area must be described significantly why authors select this area  Have been added in Location and Time of Sampling (paragraph 2)

8 Authors include the station id in the map  Have been added in Figure 1

9 Author can describe all pollution in single table to reduce the paper length and summarize it.  We apologize because each assessment method has different parameters and subjects so it is less effective to combine them into a single table

10 all of the figures are totally blur and must be redraw  Thanks for the suggestion, the figures has been fixed

11 Comparison table must be needed to compare their studies with others studies in the globe  A comparison of studies with research in various parts of the world has been included in the discussion and Table No.12

---

## [Decision Letter · Decision Letter 1]

29 Sep 2023

PONE-D-23-01059R1Heavy Metals in Water and Sediment of Cikijing River, Rancaekek District, West Java: Contamination Distribution and Ecological Risk AssessmentPLOS ONE

Dear Dr. Marselina,

Thank you for submitting your manuscript to PLOS ONE. After careful consideration, we feel that it has merit but does not fully meet PLOS ONE’s publication criteria as it currently stands. Therefore, we invite you to submit a revised version of the manuscript that addresses the points raised during the review process.

We look forward to receiving your revised manuscript.

Kind regards,

Xiaoshan Zhu, Ph.D.

Academic Editor

PLOS ONE

Journal Requirements:

Reviewers' comments:

Reviewer's Responses to Questions

**Comments to the Author**

1. If the authors have adequately addressed your comments raised in a previous round of review and you feel that this manuscript is now acceptable for publication, you may indicate that here to bypass the “Comments to the Author” section, enter your conflict of interest statement in the “Confidential to Editor” section, and submit your "Accept" recommendation.

Reviewer #1: All comments have been addressed

Reviewer #3: (No Response)

2. Is the manuscript technically sound, and do the data support the conclusions?

Reviewer #1: Yes

Reviewer #3: Partly

3. Has the statistical analysis been performed appropriately and rigorously? 

Reviewer #1: Yes

Reviewer #3: No

4. Have the authors made all data underlying the findings in their manuscript fully available?

Reviewer #1: Yes

Reviewer #3: No

5. Is the manuscript presented in an intelligible fashion and written in standard English?

Reviewer #1: Yes

Reviewer #3: No

6. Review Comments to the Author

Reviewer #1: The authors has taken the serious effort to revise the manuscript. Now the paper has improved significantly. Hence I recommend the article to Accept.

Reviewer #3: In the manuscript provided by Marselina and Wijaya, contamination dstribution and ecological risk of heavy metals have been assessed in water and sediment of Cikijing River, Rancaekek District, West Java. In my opinion, the topic addressed is interesting and important. In conclusion, the manuscript can be accepted with major amendments at PLOS ONE.

- English writing needs further polish.

- The term of heavy metals is now deprecated and potentially toxic elements should be used instead.

- The "conclusion" section in the abstract part must be improved.

- Adverse health effects of analyzed elements must be presented in the introduction section. In so doing, it is suggested that the following articles be used as a reference:

i) Marine Pollution Bulletin, 123(1-2): 34-38 (2017).

ii) Environmental Science and Pollution Research, 27(12): 13301-13314 (2020).

- The results of 'one way ANOVA' and 'one-sample t-test' analysis must be presented and discussed to comparing the mean levels of studied PTEs between the sampling sites and also between the mean levels of PTEs and maximum permissible concentrations (MPCs).

- The correlation matrix between the mean content of analyzed elements and physicochemical parameters of sediment and water samples must be provided.

- Are the reference values of PTEs for your country?

- The values of EF and Nemrow indices must be presented and clearly discussed.

- Quality of the discussion section must be improved. In so doing, the authors must be organized the discussion from the general to the specific, linking your findings to the literature, then to theory, then practice and avoid repetition from the introduction.

- For numbers in text and tables < 1.00, use three digits beyond the decimal point; for numbers between 1.00 and 9.99 use two digits beyond the decimal point; for numbers between 10.0 and 99.9, use one digit beyond the decimal point; and for concentrations ≥ 100, use the nearest whole number.

- Point sign must be used as decimal.

- The values of SD and CV(%) of analyzed elements must be presented.

- The authors should include the following references to be added in the introduction section:

iii) Bulletin of Environmental Contamination and Toxicology, 109(6): 1142-1149 (2022).

7. PLOS authors have the option to publish the peer review history of their article (what does this mean?). If published, this will include your full peer review and any attached files.

Reviewer #1: **Yes: **Yalcin TEPE

Reviewer #3: No

---

## [Author Response · Author response to Decision Letter 1]

2 Nov 2023

No Comments Response from Author

1 Adverse health effects of analyzed elements must be presented in the introduction section.

In so doing, it is suggested that the following articles be used as a reference:

i) Marine Pollution Bulletin, 123(1-2): 34-38 (2017). 13 Has been added in Introduction section

2 ii) Environmental Science and Pollution Research, 27(12): 13301-13314 (2020)  Has been added in Introduction section

3 Bulletin of Environmental Contamination and Toxicology, 109(6): 1142-1149 (2022)  Has been added in Introduction section

4 The results of 'one way ANOVA' and 'one-sample t-test' analysis must be presented and discussed to comparing the mean levels of studied PTEs between the sampling sites and also between the mean levels of PTEs and maximum permissible concentrations (MPCs)  One way ANOVA in this research is used to determine whether parameter data is homogeneous or not to choose the correlation method used. It has been added in the discussion section.

5 The correlation matrix between the mean content of analyzed elements and physicochemical parameters of sediment and water samples  It has been added in the discussion section.

6 The reference values of PTEs in Indonesia  Potential Toxic Elements for the heavy metals analyzed have not yet been regulated in Indonesia

7 The values of EF and Nemrow indices must be presented and clearly discussed  Based on other journal references, the EF and Newrow methods cannot be used in this analysis because the three methods used, namely Igeo, PLI, and PERI, can already present heavy metal contamination in ecology.

8 The values of SD and CV(%) of analyzed elements must be presented. Point sign must be used as decimal. Improved number writing.  It has been corrected and added to the manuscript

---

## [Editor Report · Decision Letter 2]

7 Nov 2023

Heavy Metals in Water and Sediment of Cikijing River, Rancaekek District, West Java: Contamination Distribution and Ecological Risk Assessment

PONE-D-23-01059R2

Dear Dr. Marselina,

We’re pleased to inform you that your manuscript has been judged scientifically suitable for publication and will be formally accepted for publication once it meets all outstanding technical requirements.

Kind regards,

Xiaoshan Zhu, Ph.D.

Academic Editor

PLOS ONE
---

## [Editor Report · Acceptance letter]

28 Nov 2023

PONE-D-23-01059R2 

Heavy Metals in Water and Sediment of Cikijing River, Rancaekek District, West Java: Contamination Distribution and Ecological Risk Assessment 

Dear Dr. Marselina:

I'm pleased to inform you that your manuscript has been deemed suitable for publication in PLOS ONE. Congratulations! Your manuscript is now with our production department. 

Kind regards, 

on behalf of

Dr. Xiaoshan Zhu 

Academic Editor

PLOS ONE